# Non-productive angiogenesis disassembles Aß plaque-associated blood vessels

Maria I. Alvarez-Vergara[1,16], Alicia E. Rosales-Nieves [1,16], Rosana March-Diaz[1], Guiomar Rodriguez-Perinan[1], Nieves Lara-Ureña[1], Clara Ortega-de San Luis [1,14], Manuel A. Sanchez-Garcia [1,15], Miguel Martin-Bornez[1], Pedro Gómez-Gálvez [1,2,3], Pablo Vicente-Munuera [1,2,3], Beatriz Fernandez-Gomez[4], Miguel A. Marchena[4,5], Andrea S. Bullones-Bolanos[1], Jose C. Davila[3,6], Rocio Gonzalez-Martinez[7], Jose L. Trillo-Contreras[1,8], Ana C. Sanchez-Hidalgo[1,8], Raquel del Toro [1,8,9], Francisco G. Scholl[1,8], Eloisa Herrera [7], Martin Trepel[10], Jakob Körbelin [11], Luis M. Escudero [1,2,3], Javier Villadiego [1,3,8], Miriam Echevarria [1,3,8], Fernando de Castro [4], Antonia Gutierrez [3,6], Alberto Rabano[12], Javier Vitorica [1,3,13] & Alberto Pascual [1✉]

The human Alzheimer's disease (AD) brain accumulates angiogenic markers but paradoxically, the cerebral microvasculature is reduced around Aß plaques. Here we demonstrate that angiogenesis is started near Aß plaques in both AD mouse models and human AD samples. However, endothelial cells express the molecular signature of non-productive angiogenesis (NPA) and accumulate, around Aß plaques, a tip cell marker and IB4 reactive vascular anomalies with reduced NOTCH activity. Notably, NPA induction by endothelial loss of presenilin, whose mutations cause familial AD and which activity has been shown to decrease with age, produced a similar vascular phenotype in the absence of Aß pathology. We also show that Aß plaque-associated NPA locally disassembles blood vessels, leaving behind vascular scars, and that microglial phagocytosis contributes to the local loss of endothelial cells. These results define the role of NPA and microglia in local blood vessel disassembly and highlight the vascular component of presenilin loss of function in AD.

[1] Instituto de Biomedicina de Sevilla (IBiS), Hospital Universitario Virgen del Rocio/CSIC/Universidad de Sevilla, Seville, Spain. [2] Department of Biología Celular, Universidad de Sevilla, Seville, Spain. [3] Centro de Investigacion Biomedica en Red sobre Enfermedades Neurodegenerativas (CIBERNED), Madrid, Spain. [4] Grupo de Neurobiología del Desarrollo-GNDe, Instituto Cajal-CSIC, Madrid, Spain. [5] Departamento de Medicina, Facultad de Ciencias, Biomédicas y de la Salud, Universidad Europea de Madrid, Villaviciosa de Odón, Spain. [6] Department of Biologia Celular, Genetica y Fisiologia, Facultad de Ciencias, Instituto de Investigacion Biomedica de Malaga (IBIMA), Universidad de Malaga, Malaga, Spain. [7] Instituto de Neurociencias de Alicante, Consejo Superior de Investigaciones Científicas-Universidad Miguel Hernández (CSIC-UMH), Alicante, Spain. [8] Department of Fisiología Médica y Biofisica, Universidad de Sevilla, Seville, Spain. [9] Centro de Investigacion Biomedica en Red de Enfermedades Cardiovasculares (CIBER-CV), Madrid, Spain. [10] Augsburg Medical Center, Department of Hematology and Oncology, Augsburg, Germany. [11] Section of Pneumology, Department of Oncology, Hematology and Stem Cell Transplantation, University Medical Center Hamburg-Eppendorf, Hamburg, Germany. [12] Fundacion CIEN, Madrid, Spain. [13] Department of Bioquimica y Biologia Molecular, Facultad de Farmacia, Universidad de Sevilla, Seville, Spain. [14] Present address: School of Biochemistry and Immunology, Trinity Biomedical Sciences Institute, Trinity College of Dublin, D2, Dublin, Ireland. [15] Present address: Centre for Inflammation Research, Queen's Medical Research Institute, University of Edinburgh, Edinburgh, UK. [16] These authors contributed equally: Maria I. Alvarez-Vergara, Alicia E. Rosales-Nieves. ✉email: apascual-ibis@us.es

Senile plaques are mainly composed of aggregated forms of the amyloid ß (Aß) peptide, which is generated by the sequential action of two proteases, the ß-secretase/BACE1 and the γ-secretase complex —whose catalytic subunit is Presenilin (PSEN1 or PSEN2)— on the Aß precursor protein (APP)[1]. Most of Alzheimer's disease (AD) cases are sporadic AD (sAD), whereas only a small percentage of patients have familiar AD (fAD) with an earlier onset and a more aggressive clinical course[2]. fAD-associated mutations, either in the *APP, PSEN1*, or *PSEN2* loci, correlated with either higher production of toxic $A\beta_{1-42}$ species (*APP*) or a higher ratio of $A\beta_{1-42}/A\beta_{1-40}$ (*PSEN*)[2]. Intriguingly, *PSEN* fAD variants are loss-of-function mutations with reduced processing of other substrates like NOTCH[3,4]. In sAD cases, age is the greatest risk factor[5] and associates with decreased γ-secretase activity over NOTCH[6].

NOTCH signaling is involved in many different cellular processes, including the growing of new vessels by angiogenesis[7–10] and the generation and maintenance of the blood–brain barrier (BBB)[11]. Remarkably, Aß plaques accumulate angiogenic/hypoxic markers but paradoxically cerebral microvasculature is rather decreased (for recent reviews see refs. [12,13]), specially around Aß deposits in the human AD brain[14–18] and in AD mice[19–21], suggesting an abnormal angiogenic process in the disease. Physiological angiogenesis is started by the hypoxia/inflammation-mediated induction of the vascular endothelial growth factor (VEGF), which binds to VEGFR2 in endothelial cells[22]. Upon ligand binding, VEGFR activates the extrusion of tip cells from the capillaries by the mobilization of the extracellular matrix, loss of the BBB, and emission of filopodia[22]. In turn, tip cells induce lateral inhibition of the neighboring stalk cells, which includes repression of VEGFR2 expression and activation of proliferation, lumen formation, and BBB genetic program[7–10]. Molecularly, lateral inhibition is mediated by the activation of Delta-like 4 (DLL4) in tip cells that, in turn, signal over NOTCH transmembrane receptors expressed by the adjacent cells, in a process requiring the activity of the γ-secretase complex[7–10]. Even partial inhibition of the DLL4/NOTCH pathway induces the initiation of pathologic angiogenesis that disassembles mature blood vessels into non-conducting tip cells, in a process termed non-productive angiogenesis (NPA)[7].

We hypothesize that a failure in lateral inhibition during angiogenesis could explain the accumulation of angiogenic markers and the reduction in Aß plaques-associated blood vessels observed in the human AD brain. To test our hypothesis, we first describe that angiogenesis initiates around Aß deposits both in the human AD brain and in an AD mouse model. Then, we show that the molecular signature of NPA is highly enriched in AD endothelial cells, abnormal vascular structures accumulate around Aß plaques in two different AD mouse models using histologic NPA markers, and the transcriptional activity of NOTCH is reduced. We also demonstrate, in mouse models, that adult genetic reduction of lateral inhibition in cerebral endothelial cells is sufficient to produce similar vascular anomalies in the absence of Aß overexpression. These abnormal vascular areas in the brain of AD mouse models replace blood vessels forming vascular scars (VaS). The disassembly of Aß plaque-associated blood vessels involves microglial recruitment and phagocytosis of endothelial cells and, again, induction of NPA is sufficient to induce vessel phagocytosis by microglia in the absence of Aß deposition.

## Results

**Angiogenesis is initiated around Aß plaques in the AD brain.** The accumulation of an extracellular proteinaceous deposit (Aß plaques) could disrupt the even distribution of cerebral capillaries, producing mild hypoxia (Fig. 1a). Thus, we first evaluated if Aß plaques induce local hypoxia in normoxic AD mouse models, using the hypoxic marker pimonidazole hydrochloride (Hypoxyprobe-1) combined with Thioflavin-S (Thio-S) staining. Remarkably, pimonidazole immunoreactivity localized around Thio-S reactive ($^+$) dense-core amyloid plaques (Fig. 1b). Interestingly, our recent work has shown that Aß plaque-associated microglia (AßAM) gene expression is characterized by a robust hypoxia inducible factor 1 (HIF1)-mediated hypoxic response[23], confirming that Aß plaques are hypoxic. Under low oxygen levels, the growth of new central nervous system vessels is normally instructed by the expression of VEGF in astrocytes[24] (Fig. 1a). Several reports have shown that VEGF is upregulated in the human AD brain[25–27] and that VEGF protein localizes within Aß plaques of AD mice[28]. However, whether VEGF induces angiogenesis in this context is under debate[29]. Thus, we studied if Aß plaque-associated astrocytes contribute to the local expression of VEGF. We combined in situ hybridization (ISH) with immunofluorescence for either the astrocytic marker glial fibrillary acidic protein (GFAP) or the ionized calcium binding adaptor molecule 1 (IBA1) microglial marker, another cell type commonly associated with Aß plaques. *Vegfa* mRNA expression was mainly associated with astrocytes (Fig. 1c–e), scarcely observed in microglial cells (Fig. 1e), and correlated with the protrusion of filopodia from nearby vessels (Supplementary Fig. 1a), suggesting angiogenic activity.

VEGF expression induces the conversion of endothelial cells (phalanx cells) into tip cells that guide the formation of a new vascular branch (Fig. 1f). To further evaluate the angiogenesis in the AD mouse brain, we studied the expression of the integrin αvß3 (Iαvß3), a transient marker of angiogenic cells[30] required for the stabilization of VEGFR2 upon binding to VEGF[31]. A previous report described that Iαvß3$^+$ cells accumulated in the human AD brain[32]. We found that Iαvß3$^+$ cells concentrated around Aß plaques in comparison with wild-type (WT) and non-Aß plaques brain areas in AD mouse models (Fig. 1g). Of note, no differences were found between WT and distal Aß plaques areas (Fig. 1g). Interestingly, the Iαvß3$^+$ cells looked qualitatively different around Aß plaques, suggesting vascular remodeling (Fig. 1g).

To extend our observations to the human AD brain, we used human samples obtained under tightly controlled conditions from AD (Braak tau pathology stages IV–VI) and age-matched Braak stage 0–I samples (Supplementary Data 1)[33]. Combination of Thio-S staining with Iαvß3 immunodetection suggested a connection between Iαvß3$^+$ cells and Aß plaques (Fig. 2a). To confirm the association between angiogenesis and Aß deposits, we first localized the position of all the Iαvß3$^+$ cells and Aß plaques (Fig. 2b and Supplementary Fig. 1b, 655 Aß plaques and 3209 Iαvß3$^+$ cells from five AD cases) and measured the load of the neurofibrillary tangles, another AD hallmark, in each AD case studied. An almost perfect direct correlation was observed between the number of Iαvß3$^+$ cells and Aß plaques (Fig. 2c) and, on the contrary, no association was found between Iαvß3$^+$ cells and tangles (Supplementary Fig. 1c). As described before[32], the number of Iαvß3$^+$ cells in control samples was very low (55 Iαvß3$^+$ cells from five control samples; Fig. 2c). Second, we performed 500 simulations where the position of Aß plaques was conserved and the location of the Iαvß3$^+$ cells was randomly generated in each AD case. We then measured the shortest geodesic distance between each Iαvß3$^+$ cell and the closest Aß plaque in each random simulation and in the experimental cases. Notably, in the five AD cases studied, the experimental shortest distance between Iαvß3$^+$ cells and Aß plaques was always significantly smaller than in the random simulations (Fig. 2d and Supplementary Fig. 1d) and, globally, the shortest distance between Iαvß3$^+$ cells and Aß plaques was also significantly

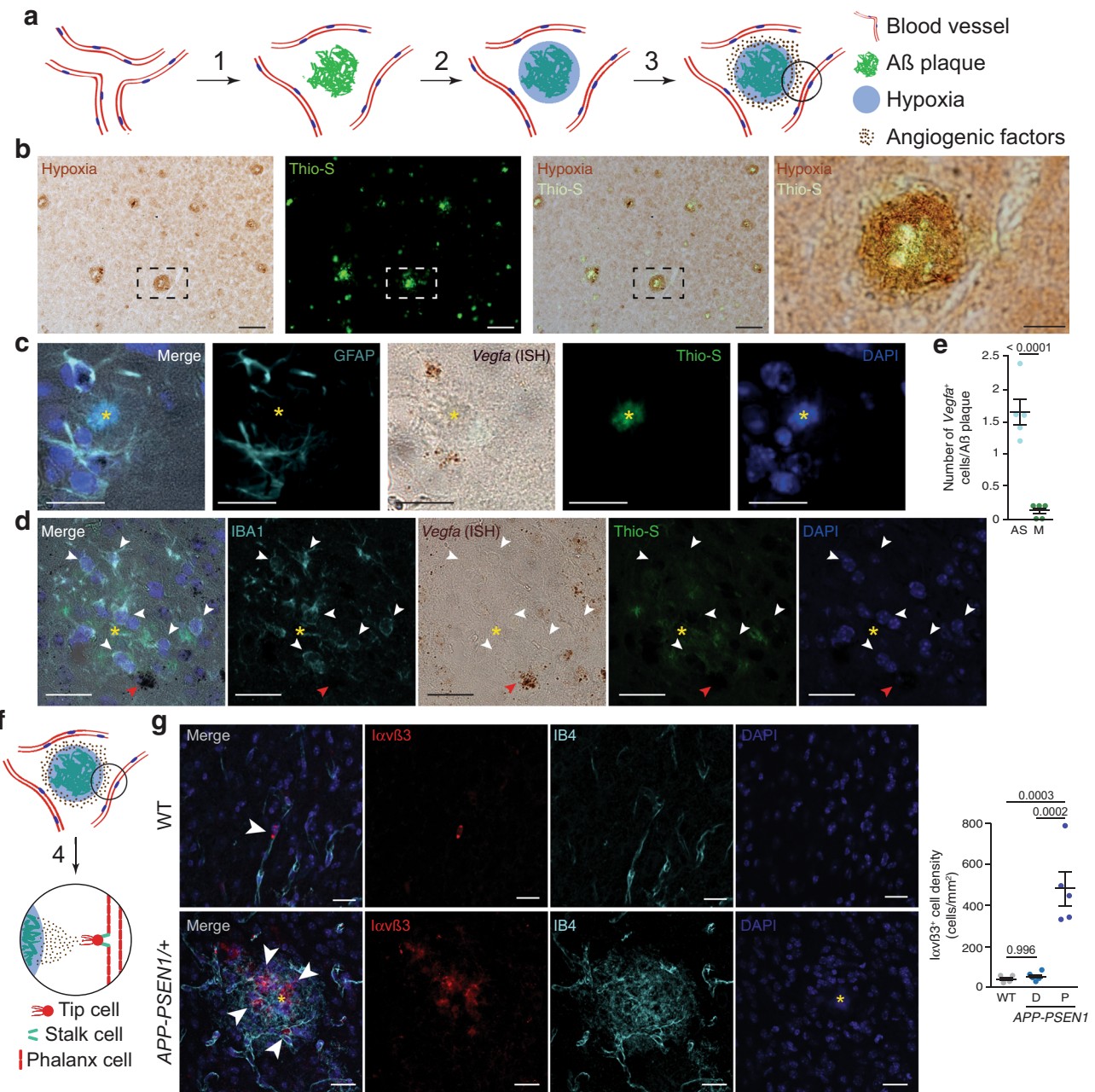

**Fig. 1 Angiogenesis is concentrated around Aß plaques in AD mouse models. a** Working model of the angiogenesis around Aß plaques. Aß deposition separate vessels (1) producing local hypoxia (2) and inducing angiogenic factors expression (3). **b** Coronal cortical sections of 14-month-old *APP-PSEN1/+* mice treated with Hypoxiprobe (Pimonidazole HCl; 60 mg/kg i.p.; 45 min) showing hypoxia (brown, immunoperoxidase, DAB) in the vicinity of Aß plaques (green, Thioflavin-S staining –Thio-S–). The dashed square box is shown in the rightest panel. Scale bar = 100 and 25 μm, respectively, in low and high magnification images. **c, d** *Vegfa* is mainly expressed by astrocytes around Aß plaques in 8-month-old *APP-PSEN1/+* mice. Aß plaques are indicated by a yellow asterisk. Cortical confocal XY images stained with astrocytic (GFAP; cyan; **c**), *Vegfa* (in situ hybridization, ISH; brown), Aß (Thio-S; green), microglial (IBA1; cyan, **d**), and nuclear (DAPI; blue) markers. White arrowheads indicate microglial cells without *Vegfa* expression and red arrowheads point to a non-microglial *Vegfa*-expressing cell (**d**). Scale bars (**c, d**) = 20 μm. **e** Quantification of the number of astrocytes (AS) and microglia (M) expressing *Vegfa* mRNA per Aß plaque. Mean ± SEM. n = 5 mice (5 Aß plaques per mice); Student's *t*-test. **f** VEGF differentiates phalanx cells to tip cells that extrude from the vessel and stalk cells that will produce the new capillary (4). **g** A cortical mouse brain area stained with angiogenic endothelial (Integrin αvß3 –Iαvß3–; red), microglial (IBA1; green), endothelial (IB4; white), and nuclear (DAPI; blue) markers. Scale bar = 20 μm. Right graph, quantification of the Iαvß3+ cell density in 8-month-old wild-type (WT) and *APP-PSEN1/+* mice. Mean ± SEM. n = 4 WT and 5 *APP-PSEN1/+* mice; ANOVA, post hoc Tukey's test.

smaller to that expected in a random distribution of Iαvß3+ cells (Fig. 2e).

Altogether, the presence of hypoxia around Aß plaques, the local induction of VEGF expression by astrocytes, and the concentration of angiogenic cells near Aß plaques strongly suggest that angiogenesis is initiated around Aß deposits in both the human AD brain and in AD mouse models.

**The angiogenesis around Aß plaques is non-productive.** As discussed before, a puzzling characteristic of the AD brain is the

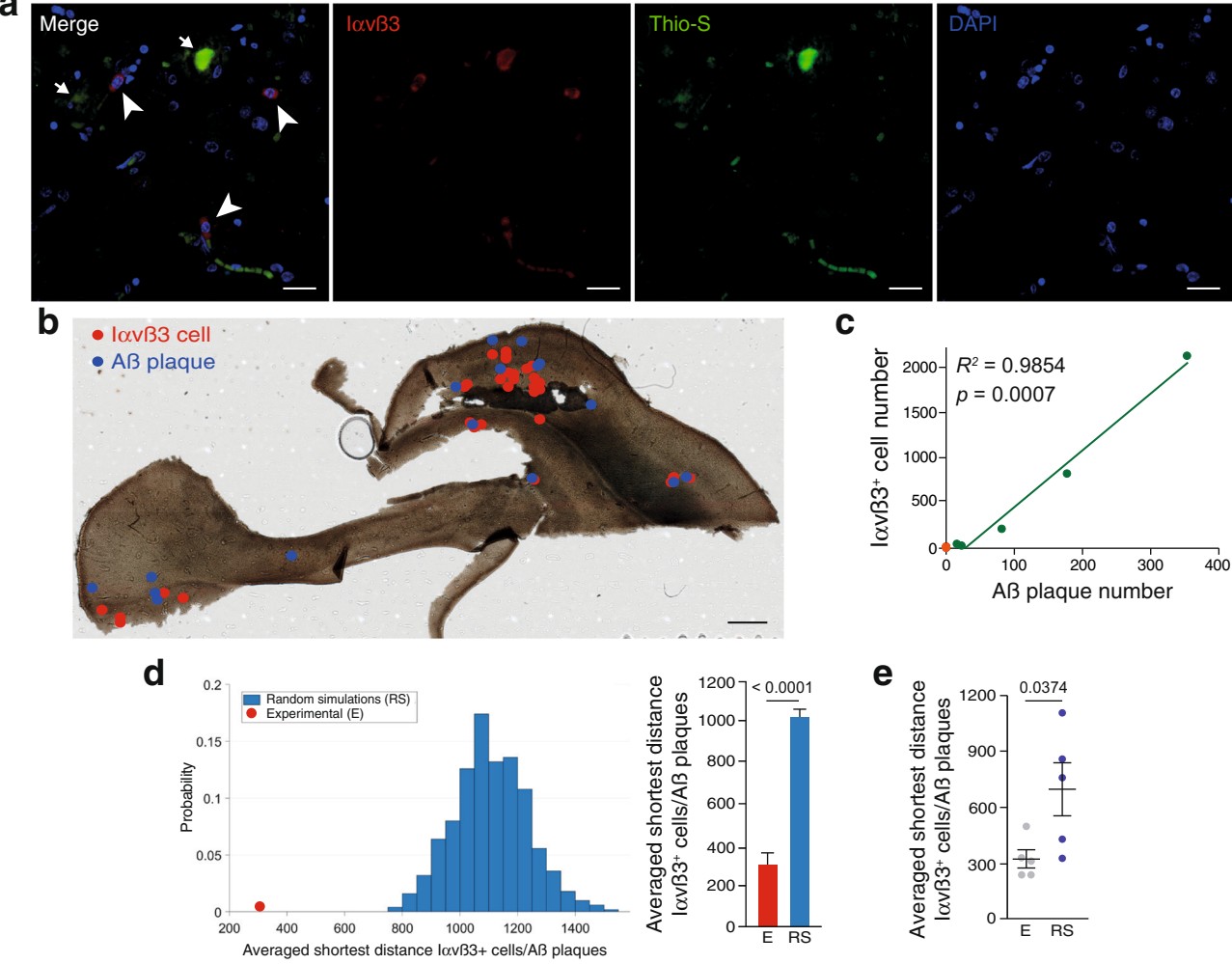

**Fig. 2 Angiogenesis is concentrated around Aß plaques in AD. a** A human hippocampal brain slice stained with angiogenic endothelial (Iαvß3; red), Aß (Thio-S; green), and nuclear (DAPI; blue) markers. Scale bar = 20 µm. Arrowheads indicate angiogenic cells and arrows signal Aß plaques. **b** A human hippocampal brain slice where the position of Iαvß3+ cells (red dots) and Aß plaques (blue dots) are indicated. Scale bar = 1 mm. **c** Correlation between Iαvß3+ cells and the number of Aß plaques in five AD (Braak IV–VI) cases (green dots). The number of Iαvß3+ cells is also indicated for six control (Braak 0–I) human samples (orange dots). Spearman r correlation. **d** Left graph, representation of the probability of the averaged shortest distance between Iαvß3+ cells and Aß plaques in 500 random simulations (RS, blue bars) where the Aß plaques position was fixed and the Iαvß3+ cells' location was randomized. The red dot represents the experimental measurement. Right graph, quantification of the shortest geodesic distance between Iαvß3+ cells and Aß plaques in an experimental (E) and in the first 10 random simulations (RS). Data are presented as mean ± SEM. n = 28 (E) and 280 (RS) Iαvß3+ cells and 16 Aß plaques. Student's t-test. **e** Quantification of the averaged shortest geodesic distance between Iαvß3+ cells and Aß plaques from experimental (E) human brain slices and 500 random simulations (RS) of Iαvß3+ cells location. Mean ± SEM. n = 5 human samples; Student's t-test.

accumulation of angiogenic markers coupled with reduced number of vessel and disruption of the BBB. To investigate if angiogenesis is halted by a failure in differentiation between tip and stalk cells (NPA) in AD mouse models (Fig. 3a), we separated CD31+ (an endothelial/innate immune cell marker) and CD11b-negative (an innate immune marker) cells using fluorescence-activated cell sorting (Supplementary Fig. 2a–c) from aged (18-month-old) *APP-PSEN1/+* and WT mice. mRNA levels of a vascular specific marker (*Cadherin 5, Cdh5*) greatly exceeded the levels of microglia, astrocyte, oligodendrocyte, and neuronal markers (Supplementary Fig. 2d), confirming the purity of the isolated endothelial cells. A global expression analysis was performed using microarrays and the differentially expressed (DE) genes are shown in the Supplementary Data 2. To evaluate whether endothelial cells from an Aß-accumulating AD mouse model could suffer NPA, we defined a gene set (GS) containing

the genes upregulated in the retina of *Dll4* heterozygous mice (Dll4+/−_Up)[34], a model of NPA[7–10], and estimated its contribution to the DE genes between *APP-PSEN1/+* and WT endothelial cells using GS enrichment analysis (GSEA). Notably, the Dll4+/−_Up GS was enriched at the top of the list —and the only one with a significant FWER *p* value— out of more than 800 GS analyzed (Fig. 3b, c and Supplementary Data 3), strongly suggesting a failure of the endothelial lateral inhibition in AD mouse models. NPA induces an increase in the number of tip cells in the retina of mouse models[7–10]. From the list of the DE genes included in the Dll4+/−_Up GS, we selected the *Plasminogen activator, urokinase receptor* (*Plaur*) gene, a tip cell marker[34], for histological validation of our molecular studies. We combined ISH with isolectin B4 (IB4) from *Griffonia simplicifolia* staining, a well-described marker of mouse mature and angiogenic endothelial cells[35]. Low expression of *Plaur*

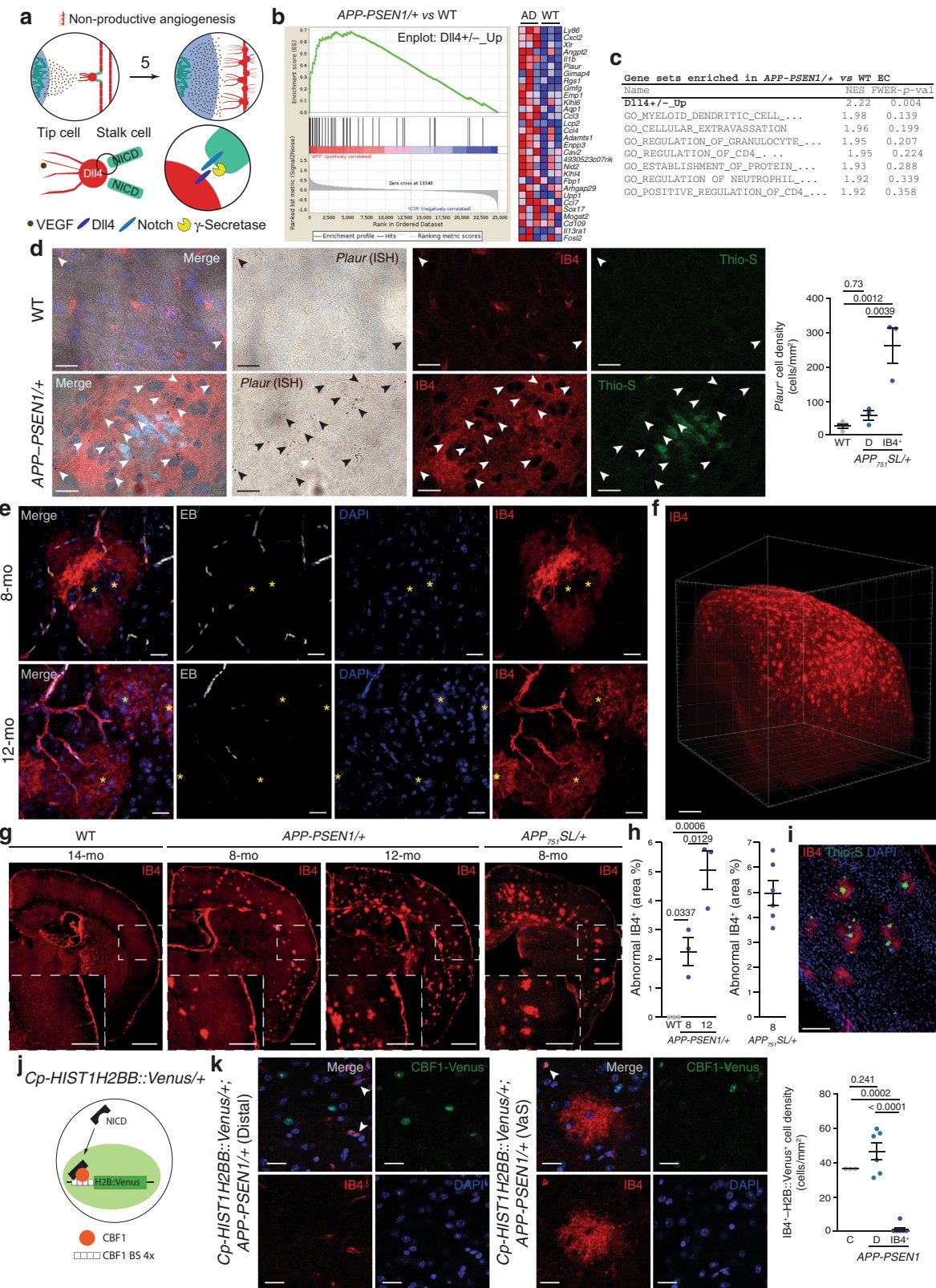

mRNA in endothelial cells was observed in WT or in regions distal to Aß plaques in an AD mouse model (Fig. 3d). However, a dramatic increase in *Plaur* mRNA expression was observed around Aß plaques (Fig. 3d), further demonstrating that, although the tip cell program is activated, the DLL4/NOTCH interaction (see Fig. 3a) that limits tip cell accumulation is affected near the Aß plaques.

IB4 is also a marker of the vascular abnormalities observed in NPA models[7–10] characterized by the accumulation of tip cells (Fig. 3a). As expected, IB4 delineated the vessels distal to Aß plaques (Fig. 3e and Supplementary Fig. 3a); however, IB4 recognized abnormal structures around Aß deposits. Those structures presented a "cotton candy"-like appearance, and emanated from well-perfused IB4+ vessels (Evans blue —EB— angiography; Fig. 3e and

**Fig. 3 Angiogenesis is non-productive around Aß plaques. a** Working model of the angiogenesis around Aß plaques. Numeration continues from Fig. 1f. Upper row: Non-productive angiogenesis will convert phalanx cells to non-conducting tip cells, extending local hypoxia (5). Lower row: γ-secretase activity is involved in the lateral inhibition process that controls tip-stalk cell identity. **b, c** Gene set enrichment analysis (GSEA) revealed that Dll4/–_Up GS is highly represented in 18-month-old APP-PSEN1/+ versus WT endothelial cell differential transcriptomic (**b**, left panel). Right panel (**b**) shows the heat map of the top 30 ranking leading edge genes included in the Dll4/–_Up GS. Red symbolizes overexpression and blue down regulation. The table includes the eight top-enriched GSs (**c**), FEWER p val (values) were two-sided and adjusted for multiple comparisons. **d** Cortical confocal XY images from 8-month-old APP-PSEN1/+ and stained with endothelial (IB4; red), Plaur (ISH; brown), Aß (Thio-S; green), and nuclear (DAPI; blue) markers. Arrowheads indicate reactive cells expressing Plaur. Scale bar = 20 μm. Right graph shows the quantification of Plaur+ cells/mm2 in WT (gray dots) and distal to Aß plaques (D; light blue dots) and IB4+ regions (IB4+; blue dots) in the APP-PSEN1/+ mouse model. Mean ± SEM. n = 4 WT and 3 APP-PSEN1/+ mice; ANOVA, post hoc Tukey's test. **e** Cortical confocal projection from 8- (upper row) and 12- (lower row) month-old APP-PSEN1/+ mice injected with Evans Blue (EB, white) and stained with endothelial (IB4; red) and nuclear (DAPI; blue) markers. Aß plaques are indicated with a yellow asterisk. Scale bar = 20 μm. **f** Full hemi-cortex from a 10-month-old APP-PSEN1/+ mouse stained to visualize endothelial cells (IB4; red) and rendered transparent using iDISCO. Scale bar = 500 μm. **g** Superimages of brain cortical sections from WT, APP-PSEN1/+, and APP751SL/+ mice stained to label endothelial cells (IB4; red). Insets show the white square from low magnification images. Scale bar = 1 mm in low and 500 μm in high magnification images. **h** Quantification of the percentage of cortical surface occupied by abnormal IB4+ staining. Mean ± SEM. n = 3 8-month-old WT and APP-PSEN1/+; and 12-month-old APP-PSEN1/+; and 6 APP751SL/+ mice; ANOVA, post hoc Tukey's test. **i** Image of a cortical slice from 8-month-old APP-PSEN1/+ mice stained with endothelial (IB4; red), Aß (Thio-S; green), and nuclear (DAPI; blue) markers. Scale bar = 100 μm. **j** Schematic representation of the Cp-HIST1H2BB::Venus/+ mouse model. NICD NOTCH intracellular domain, CBF1 BS CBF1-binding sites. **k** Left images: coronal cortical sections from Cp-HIST1H2BB::Venus/+; APP-PSEN1/+ mice distal (left) and proximal (right) to Aß plaques and stained with endothelial (IB4; red), and nuclear (DAPI; blue) markers. Green: Direct visualization of H2BB::Venus fluorescence. Scale bar = 20 μm. Right graph, quantification of the number of H2BB::Venus positive endothelial cells in Cp-HIST1H2BB::Venus/+; +/+ (Control, C), Cp-HIST1H2BB::Venus/+; APP-PSEN1/+ distal (D) and IB4+ proximal (P) to Aß plaques. Mean ± SEM. n = 3 Cp-HIST1H2BB::Venus/+; +/+ and 6 Cp-HIST1H2BB::Venus/+; APP-PSEN1/+ mice; ANOVA, post hoc Tukey's test.

Supplementary Fig. 3a). The IB4+ vascular anomalies were not colocalized either with astrocytes (Supplementary Fig. 3b) or microglial cells (Supplementary Fig. 3c), the two cell types more commonly found in the proximity of Aß plaques. To further evaluate the contribution of innate immune cells to the IB4+ vascular abnormalities, we examined the cerebral vasculature of an experimental autoimmune encephalomyelitis (EAE) mouse model[36] (Supplementary Fig. 3d), including 3 sham control and 12 EAE mice distributed in three groups based on behavioral evaluation and the time of evolution after the induction of the disease (4 onset, 4 peak, and 4 post-peak mice; Supplementary Fig. 3e). We did not find any IB4+ vascular anomalies, despite the strong glial activation observed around endothelial cells (Supplementary Fig. 3, compare panel 3f—onset—with 3g—peak; 0 IB4+ vascular abnormalities in 103 autoimmune foci examined).

In the APP-PSEN1/+ AD mouse model, the IB4+ vascular anomalies were found in regions containing Aß plaques, including the cortex, hippocampus, and the corpus callosum (Fig. 3e–g and Supplementary Movie 1), covering up to 5% of the total cortical surface (Fig. 3h) and not detected in WT mice (Fig. 3h). Although those mice express dominant APP and PSEN1 mutations in neuronal cells (driven by the Prnp promoter[37]), our molecular analysis has predicted a failure of NOTCH signaling in endothelial cells, which involves the activity of the γ-secretase (Fig. 3a). To discard any alteration caused by PSEN1 mutated allele to endothelial cells, we tested whether mutant mice expressing only the APP gene will also accumulate IB4+ vascular anomalies. The analysis of APP751SL/+, an AD model with faster Aß plaque deposition, revealed qualitative (Fig. 3g) and quantitative (Fig. 3h) similar accumulation of IB4+ vascular anomalies, excluding the possibility of endothelial expression of the mutated form of PSEN1 as the cause of the vascular alterations in AD mouse models, and strongly suggesting that Aß plaques induce vascular disorganization. To quantitatively confirm the connection between the abnormal IB4+ structures and Aß plaques, we co-stained brain slices from Aß-depositing AD mouse models with Thio-S and IB4+ (Fig. 3i). In the brain of young AD mice without or with a scarce number of Aß deposits (respectively 3-month-old and 5-month-old APP-PSEN1/+ mice), IB4+ vascular abnormalities were always colocalized with Thio-S (Supplementary Fig. 3h). In older mice, quantification of

the area of both Aß deposits (Thio-S+) and the abnormal IB4+ structures revealed a significant positive correlation between both parameters in the regions analyzed (Supplementary Fig. 3i, j).

So far, we have shown molecular and histological indications of NPA. To further demonstrate the loss of NOTCH activity around Aß plaques, we generated a new AD mouse model by crossing APP-PSEN1/+ with a NOTCH reporter mice expressing a fluorescent nuclear protein (Histone H2B fussed with Venus) under the control of CBF1-binding sites[38] (Fig. 3j). Upon ligand binding, NOTCH is cleaved by several proteases at the membrane, including a final cleavage by the γ-secretase, and the NOTCH intracellular domain (NICD) is translocated to the nucleus, where it binds to CBF1, recognizes CBF1-binding sites at the DNA, and activates transcription. We first checked that the expression of the reporter was compatible with the described activity of NOTCH in the adult brain. We focused on the neurogenic subventricular zone, where NOTCH expression and activity has been reported[39], and observed a clear nuclear Venus signal lining the ventricles, validating the model. In the cortex, we observed expression of the reporter in several cell types including the endothelium (Fig. 3k). Quantification of the endothelial expression of the NICD reporter showed reduced expression in the IB4+ vascular abnormalities when compared with WT or APP-PSEN1/+ brain areas distal to Aß plaques (Fig. 3k).

Altogether, the induction of early angiogenesis markers (VEGF and Iαvß3+), the molecular signature of NPA in vascular cells, the accumulation of vascular abnormalities around Aß plaques, and the decreased activity of NICD associated with the vascular anomalies, strongly suggest that Aß plaques are associated with NPA.

**Endothelial γ-secretase LOF induces IB4+ vascular anomalies.** Loss of function of γ-secretase has been linked both with fAD (PSEN1 and PSEN2 mutations)[2–4] and sAD (decreased NOTCH-processing activity associated with age)[6]. Due to the role of the γ-secretase in the production of Aß, many of the studies have been focused in the loss of γ-secretase in neurons[4]. However, mutations in PSEN1/PSEN2 will modify γ-secretase activity in all the cells and, during aging, other cells types may also be altered by a decrease in γ-secretase activity, and therefore, contribute to the progression of AD. In addition, it has been proposed that the high

concentration of $A\beta_{1-42}$ around A$\beta$ plaques could inhibit $\gamma$-secretase activity as a product of the reaction[4]. We therefore investigate the consequences of reducing $\gamma$-secretase activity in adult AD brain endothelial cells. To this end, we genetically inhibited endothelial $\gamma$-secretase activity by injecting $Psen1^{loxP/loxP}$; $Psen2^{-/-}$ mice with cerebral endothelium-specific Cre recombinase-expressing adeno-associated vectors (AAV-BR1; Fig. 4a)[40]. First, we validated that the viral vector induced the deletion of the $Psen1$ gene in the brain. We designed a quantitative PCR (qPCR) amplicon to detect the $Psen1$ excised locus (Fig. 4b) and verified that the injected mouse induced the loss of $Psen1$ allele (Fig. 4b) at two different time points after the injection of the viral vector, 19 and 60 days. Interestingly, a trend was observed to reduce the deletion with time (19 days, $0.33 \pm 0.12$ versus 60 days, $0.14 \pm 0.05$; Student's $t$-test: $p = 0.126$). Second, we checked if the $Psen1$ gene deletion was specific to endothelial cells. To this end, we performed $Psen1$ mRNA ISH combined with IB4 immunofluorescence (Fig. 4c and Supplementary Fig. 4a) and quantify the number of RNA foci in the whole brain parenchyma (cortex and striatum). As expected, no differences were found in the total number of $Psen1$ mRNA foci (Supplementary Fig. 4a). Thus, we quantified the number of endothelial $Psen1$ mRNA foci, observing a trend to decrease in the striatum and significant decrease in the cortex (Fig. 4d; see below for further discussion of this experiment).

To evaluate the vasculature of mice with endothelial loss of the $\gamma$-secretase activity, we used several perfusion and BBB markers at two different time points. Nineteen days after the viral injection, no evident abnormalities were found. At 60 days, a normal brain microvasculature was observed in mice injected with the control vector (Fig. 4e); however, vessels were deeply altered in the brain of mice injected with Cre-expressing AAVs (AAV-BR1-Cre; Fig. 4e). Notably, we observed structures that were identical to the IB4$^+$ vascular abnormalities observed around A$\beta$ plaques (compare Fig. 3e–i with Fig. 4e), even though the mouse model employed did not accumulate A$\beta$ deposits. In this model, IB4$^+$ vascular abnormalities were mainly found in the striatum and hippocampus (Fig. 4e) and were associated with a loss in perfusion (EB and TER119, Supplementary Fig. 4b, c), pericytes (platelet derived growth factor receptor ß, PDGFRß; Fig. 4f), and astrocytic end-feet (Aquaporin 4, AQP4; Fig. 4g). Interestingly, no signs of endothelial proliferation were found using Ki67 (53 IB4$^+$ vascular abnormalities analyzed from five mice), suggesting, as expected, that NPA is blocking the differentiation of endothelial cells to the proliferative stalk phenotype. As a control, we identified proliferative (Ki67$^+$) cells in the subgranular zone of the hippocampus and in proliferating microglia.

Altogether, our data suggest that loss of vascular cells is induced by the endothelial loss of function of $\gamma$-secretase, which leaves behind IB4$^+$ vascular abnormalities.

**Vessel loss and accumulation of VaS around A$\beta$ plaques.** The loss of blood vessels associated with the IB4$^+$ vascular abnormalities were quite reminiscent of the defects observed around A$\beta$ plaques in the human AD brain[14–18] and in AD mice[19–21]. To evaluate if A$\beta$ plaques-associated IB4$^+$ vascular abnormalities were also characterized by blood vessel loss, we analyzed perfusion in $APP_{751}SL/+$ using EB angiography[35] and quantification of marker of red blood cells, TER119 (Fig. 5a). Quantification of EB signal showed a clear reduction in perfusion proximal to A$\beta$ plaques compared with distal regions (Fig. 5a). As previous works have reported disturbed brain blood flow[12,13,26] and neutrophils clotting of the cerebral capillaries[41] in AD mouse models, we analyzed brain vessels in WT and in distal and proximal regions to A$\beta$ plaques in an AD mouse model using TER119 staining.

Interestingly, loss of perfusion was only observed around A$\beta$ plaques and no significant differences were found between wild type and distal cortical regions (Fig. 5a). To study if reduced A$\beta$ plaques perfusion was associated with a morphologic change of capillaries around A$\beta$ plaques, we used several markers of the blood–brain barrier. First, we examined the distribution of laminin—a marker of the endothelial basement membrane—combined with IB4. As expected, both laminin and IB4 delineated the vessels distal to A$\beta$ plaques (Fig. 5b); however, laminin staining was reduced in the IB4$^+$ vascular abnormalities (Fig. 5b). We quantified the area occupied by laminin$^+$ vessels in a 50 µm radius from A$\beta$ plaques and inside the abnormal IB4$^+$ area and observed a significant reduction in two different AD mouse models (Fig. 5b). Again, no differences were found between WT and distal brain regions of the AD mouse models. Similar to laminin, the expression of platelet/endothelial cell adhesion molecule 1 (PECAM1/CD31; a tight junction marker) was decreased around A$\beta$ deposits (Supplementary Fig. 5a) and no changes were observed between WT and distal regions in the AD mouse model. Pericyte number is decreased in both patients and AD models[18] and the reduction correlates with A$\beta$ deposits[17]. Correspondingly, the expression of the PDGFRß was lost in the anomalous IB4$^+$ structures (Supplementary Fig. 5b) and, similar to other vascular markers, no changes in PDGFRß expression were observed between WT and distal regions in the AD mouse model (Supplementary Fig. 5b). Finally, AQP4, a marker of astrocytic end-feet, delineated the blood vessels in regions distal to A$\beta$ plaques (Fig. 5c), but, as described[42,43], a diffuse signal was observed in the proximity of A$\beta$ deposits that colocalized with the IB4$^+$ vascular anomalies in some areas (Fig. 5c), including clearly recognizable astrocytic end-feet (arrowheads in Fig. 5c). Quantification of AQP4$^+$ signal revealed no differences between WT and distal regions in the AD mouse models and a significant reduction around A$\beta$ plaques. Altogether, our results indicate that A$\beta$ plaques are avascular areas were the vessels have been substituted by an abnormal IB4$^+$ signal.

To localize IB4 staining at the ultrastructural level, we used gold-based IB4 staining and electron microscopy. IB4 labeling was mainly restricted to the extracellular space surrounding A$\beta$ fibrils (Fig. 5d), a parenchyma that is normally invaded by microglial and astrocytic projections. Based on their continuity from perfused blood vessels (Figs. 3 and 4) and their accumulation at the extracellular space around the A$\beta$ deposits (Fig. 5), we termed these IB4$^+$ anomalies vascular scars (VaS).

To evaluate the contribution of abnormal angiogenesis to the progression of the disease, we treated an AD mouse model with sorafenib, a drug that inhibits the intracellular activity of several angiogenic kinases (VEGFR, PDGFR, and RAF) in endothelial cells. Sorafenib treatment (30 mg/kg every 2 days for 1 month) strongly reduced the accumulation of I$\alpha$v$\beta$3$^+$ cells in the brain of 8-month-old $APP_{751}SL/+$ mice (Supplementary Fig. 6a), indicating that the selected dose of Sorafenib was enough to reduce angiogenic activity. We then examined A$\beta$ deposits using Thio-S staining and revealed that Sorafenib presented a trend to reduce both A$\beta$ plaque load and the mean A$\beta$ plaque area (Supplementary Fig. 6b) without altering the $A\beta_{1-40}$, $A\beta_{1-42}$, and $A\beta_{1-42}/A\beta_{1-40}$ levels, as estimated by ELISA (Supplementary Fig. 6c). In addition, analysis of pericyte coverage of blood vessels, as a readout of vessels normalization, showed that sorafenib treatment recovered the PDGFRß signal around A$\beta$ plaques to the same levels observed in distal regions (Supplementary Fig. 6d). Finally, we estimated the short-term memory using the novel object recognition test and observed a trend to improve the memory in sorafenib-treated mice (Supplementary Fig. 6e), suggesting a modest recovery in cognition. Therefore, although

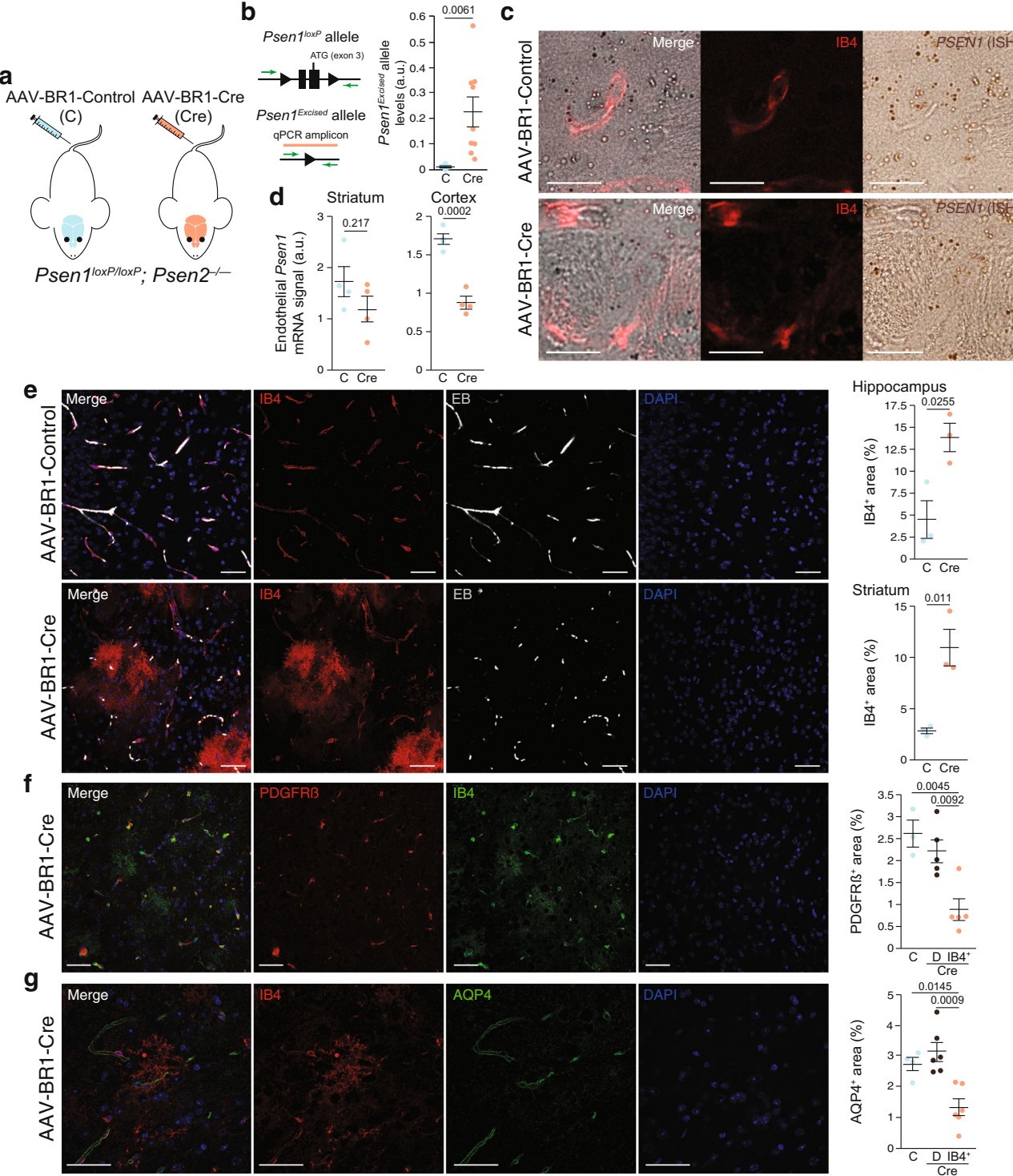

**Fig. 4 In adult inhibition of endothelial γ-secretase activity is sufficient to generate IB4$^+$ vascular abnormalities. a** Schematic representation of a mouse model with adult inhibition of endothelial γ-secretase activity. *Psen1*$^{loxP/loxP}$; *Psen2*$^{-/-}$ mice were injected with cerebral endothelium-specific adeno-associated control (AAV-BR1-Control; C) or Cre recombinase-expressing (AAV-BR1-Cre; Cre) viruses. **b** Schematic representation of the qPCR amplicon used to detect the *Psen1* excised allele (orange bar). Right graph: Quantification of the degree of *Psen1* excision (a.u., arbitrarty units) in the striatum of C and Cre mice. Mean ± SEM. *n* = 3 C and 10 Cre mice; Student's *t*-test. **c** Striatal confocal XY images from C and Cre mice stained with endothelial (IB4; red) and *Psen1* (ISH; brown) markers. Scale bar = 20 µm. **d** Quantification of endothelial *Psen1*$^+$ signal in the striatum (left graph) and the cortex (right graph) of C and Cre mice. Mean ± SEM. *n* = 4 mice; Student's *t*-test. **e–g** Confocal projections of striatal (**e**) or hippocampal slices (**f, g**) from *Psen1*$^{loxP/loxP}$; *Psen2*$^{-/-}$ mice injected with AAV-BR1-Control or AAV-BR1-Cre viral vectors, and, 2 months later, perfused with Evans blue (EB; white —**e**) and stained with endothelial (IB4; red —**e, g** or green —**f**), pericyte (PDGFRß; red —**f**), astrocytic end-feet (AQP4; green —**g**), and nuclear (DAPI; blue) markers. Scale bars = 40 µm. Right graphs are the quantification of: **e** percentage of area occupied by IB4$^+$ in C and Cre mice in hippocampus (upper row) and striatum (lower row). Mean ± SEM. *n* = 3 mice; Student's *t*-test. **f, g** Percentage of area occupied by PDGFRß$^+$ (**f**) or AQP4$^+$ (**g**) signal (in distal —D— vessels and in IB4$^+$ area) in hippocampus form C and Cre mice. Mean ± SEM. *n* **f** = 3 C and 5 Cre mice; *n* **g** = 4 C and 6 Cre mice ANOVA, post hoc Tukey's test.

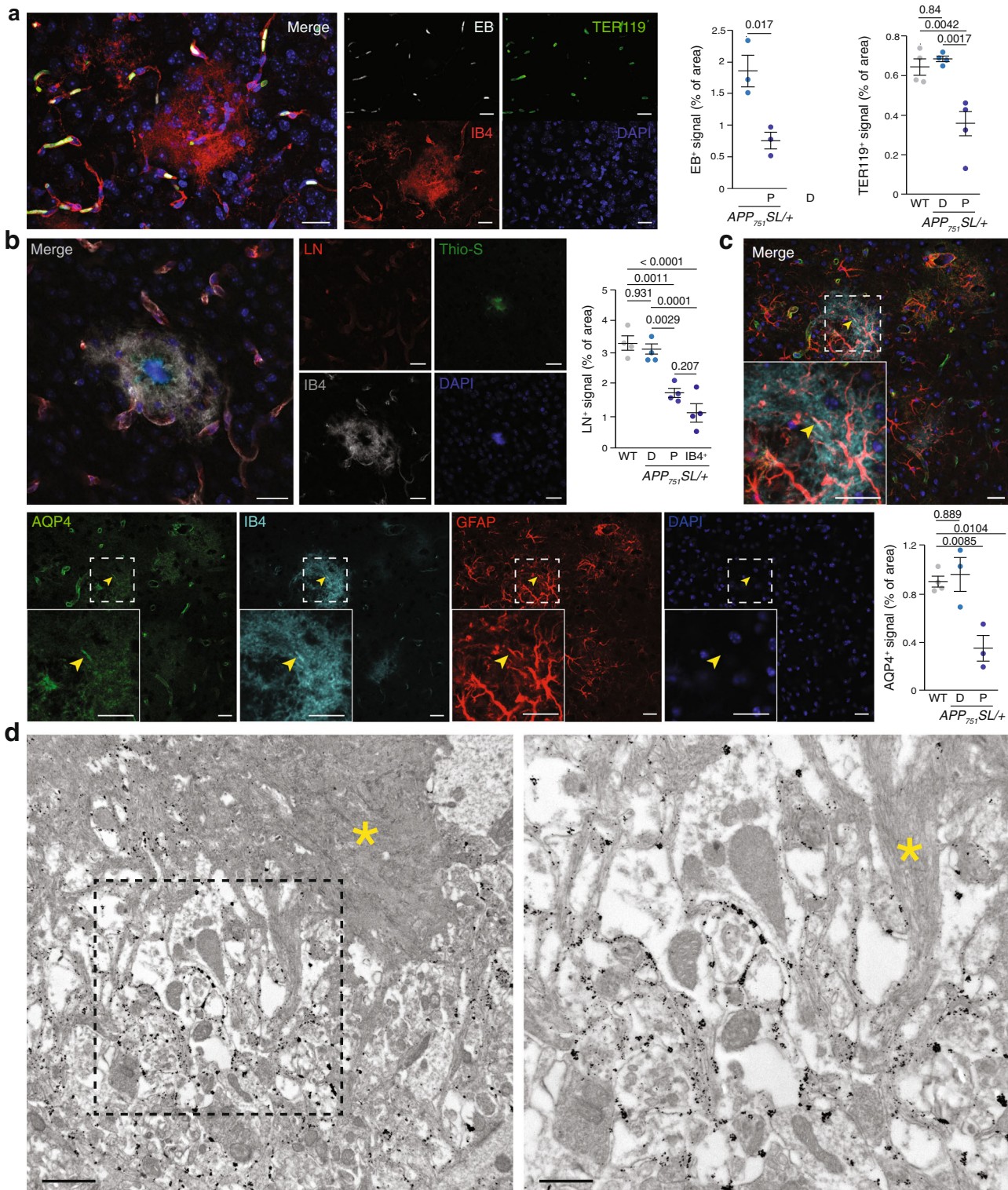

an antiangiogenic treatment may be able to delay progression of the disease, a pharmacological intervention to resume the halted angiogenic activity in AD could be more appropriated.

### NPA induces microglia phagocytosis of blood vessels near Aß plaques.

We postulated that the angiogenic cells with halted differentiation could be eliminated by the activity of other cells based on (i) the lack of mature markers of endothelial cells around Aß plaques, (ii) *Psen1* excision reached at 19 days after AAV-BR1-Cre injection was higher than the observed at 60 days, suggesting that the endothelial cells without γ-secretase activity (those without *Psen1*) could be removed, and (iii) the number of *Psen1* mRNA foci after viral Cre injection was significantly decreased in the cortex (a brain area without VaS accumulation) whereas only a trend was observed in the striatum, suggesting that in the later, the cells that lose the *Psen1* allele were either differentiated and lost the expression of IB4 or were removed. Microglia are the main phagocytic cells in the brain; therefore, we examined the microglia in the VaS

**Fig. 5 Vessels are substituted by vascular scars proximal to Aß plaques. a** Left panels, cortical confocal XY images from 8-month-old $APP_{751}SL/+$ mice stained with endothelial (IB4; red), red cells (TER119; green), Evens Blue (EB, white), and nuclear (DAPI; blue) markers. Scale bar = 20 µm. Right graphs, left, quantification of the EB vessel area distal (D) and proximal (P) to Aß plaques. Mean ± SEM. n = 3 mice; Student's t-test; right, quantification of the TER119 vessel area in WT, distal (D), and proximal (P) to Aß plaques in $APP_{751}SL/+$ mice. Mean ± SEM. n = 4 mice; ANOVA, post hoc Tukey's test. **b** Left panels, cortical confocal XY images 8-month-old $APP_{751}SL/+$ mice stained with vessel basement membrane (laminin, LN; red), endothelial (IB4; white), Aß (Thio-S; green), and nuclear (DAPI; blue) markers. Scale bar = 20 µm. Right graphs, quantification of the laminin vessel area in WT mice, in distal (D), proximal (P) to Aß plaques, and inside the IB4+ vascular abnormal structures (IB4+) in 8-month-old $APP_{751}SL/+$ mice. Mean ± SEM. n = 4 mice; ANOVA, post hoc Tukey's test. **c** Cortical confocal projection 8-month-old $APP_{751}SL/+$ mice stained with astrocytic end-feet (aquaporin 4, AQP4; green), endothelial (IB4; cyan), astrocytic (GFAP, red), and nuclear (DAPI; blue) markers. Insets show the white square from low magnification images. Yellow arrowheads indicate an astrocytic end-feet juxtaposed to an IB4+ structure. Scale bar = 20 µm. Lower graph, left, quantification of the EB vessel area distal (D) and proximal (P) to Aß plaques. Mean ± SEM. n = 4 WT and 3 $APP_{751}SL/+$ mice; ANOVA, post hoc Tukey's test. **d** Electron microscopy analysis of an 8-month-old $APP_{751}SL/+$ cortex stained with IB4 (black dots, gold particles). Right image is a high magnification of the left dashed square shown in the left panel. A yellow asterisk indicates an Aß plaque. Scale bar = 1 µm in low and 0.5 µm in high magnification images.

generated by deletion of the γ-secretase activity and in those associated with Aß plaques.

Adult inhibition of endothelial γ-secretase activity induced the formation of circular cytoplasmic microglial pouches —ball-and-chain structures characteristic of phagocytic microglia[44]— that enveloped IB4+ material (Fig. 6a and Supplementary Movies 2 and 3). Although astrocytic end-feet were decreased in VaS (Fig. 4g), astrocytic projections could also be involved in the phagocytosis of endothelial cells. However, we did not observe any abnormal vascular staining that colocalized with astrocytes (Supplementary Fig. 7a). Altogether, our data indicate that induction of NPA is sufficient to disassemble blood vessels and to induce endothelial cell phagocytosis by microglia.

To study the contribution of microglial phagocytosis to the loss of Aß plaques-associated blood vessels, we generated a new AD mouse model where tdTomato was conditionally expressed in the cytoplasm of endothelial cells (APP-PSEN1/+; Cdh5-Cre::ERT2/+; tdTomato/+) upon tamoxifen (TMX) treatment. Non-treated mice did not show any tdTomato staining and blood vessels were clearly identified in brain sections of TMX-treated WT and APP-PSEN1/+ mice (Supplementary Fig. 7b). In addition, we also observed, although with low frequency, tdTomato+ cells extruding from blood vessels (Fig. 6b and Supplementary Fig. 7c), which could represent non-terminally differentiated tip/stalk cells. Interestingly, those cells were found in close apposition to microglia (Fig. 6b and Supplementary Fig. 7c) and their projections were total or partially covered by microglial cytoplasmic extensions (Fig. 6b and Supplementary Fig. 7c). To confirm that microglia are indeed involved in the local loss of Aß plaque-associated blood vessels, we first studied the spatial distribution of microglial and endothelial cells in the absence of pathology (WT mice) and distal and proximal to Aß plaques in an AD mouse model. Low magnification images revealed that AßAM were found covering blood vessels (Fig. 6c, Supplementary Fig. 7d and Supplementary Movies 4–7), something that was not observed in WT mice or in microglia distal to Aß deposits (Supplementary Fig. 7b, d). Quantification of the length of blood vessels covered by microglia in control and distal areas from Aß plaques in an AD mouse model revealed no differences (Fig. 6c); however, a clear increase in coverage was observed around Aß plaques (Fig. 6c). Finally, we searched for phagocytic pouches similar to those found in the endothelial secretase inhibition model (Fig. 6a). tdTomato+ microglial pouches were found almost in every Aß plaque analyzed (Fig. 6d, Supplementary Fig. 7e and Supplementary Movies 8–10) and in almost all the cases (97%) they were also reactive for the lysosomal marker CD68 (Fig. 6e), indicating phagocytic activity.

Altogether, our results indicate that halted angiogenic cells, without γ-secretase activity or associated with Aß plaques, were recognized by microglia and removed by phagocytosis (Fig. 6f).

## Discussion

Vascular alterations in the AD brain have been linked with the accumulation of Aß in the wall of blood vessels in the form of cerebral amyloid angiopathy[12] and/or with a direct effect of extracellular Aß over vascular function[45]. However, both the human AD and AD mouse models' brains (1) show a reduction in the number of vessels and a debilitation of the BBB around Aß plaques[14–21] and (2) accumulate hypoxic/angiogenic markers around Aß deposits (see ref. [23] and for recent reviews see[12,13]), suggesting an important role of Aß plaques in the continuum of AD cerebral microvasculature dysfunction and in the induction of compensatory angiogenesis. We show here that, although angiogenesis is initiated around Aß plaques, the process is non-productive leading to the disassembly of Aß plaque-associated blood vessels and the phagocytic activity of microglia.

The accumulation of extracellular Aß in plaques could alter the even distribution of brain capillaries producing, together with the recruitment of innate immune cells, hypoxia-mediated VEGF expression. Indeed, hypoxia and VEGF accumulates around Aß plaques[23,25–28] and, while some authors proposed that VEGF could be sequestered in Aß plaques being biologically unavailable[28,29], we observed a high expression of VEGF in astrocytes surrounding the Aß plaques and an association between VEGF expression and the protrusion of filopodia from endothelial cells in mouse models. Notably, we show that the expression of the angiogenic Iαvß3, an endothelial cell surface glycoprotein complex and characteristic of VEGF-stimulated vessels[30,46], is associated with Aß plaques, indicating that angiogenesis is also initiated in the human AD brain around these deposits.

However, we show here that angiogenesis is non-productive around Aß plaques, (i) the molecular signature of defective lateral inhibition[34] is enriched in endothelial cells isolated from an AD mouse model, and (ii) two NPA marker (IB4 and Plaur mRNA)[7–10,34] accumulate in VaS. IB4 is a lectin that binds to glycoproteins both at the luminal and abluminal sides of endothelial cells and, in the later, it is associated with the basement membrane that surrounds mature endothelial cells and pericytes[35]. During angiogenesis, IB4 is the best histological marker of tip cells, labeling the full length of angiogenic filopodia[35]. Plaur is one of the few genes strongly enriched in tip cells versus other endothelial cells[34] and is involved in the degradation of the extracellular matrix during angiogenesis[47]. IB4+ VaS are probably the result of the angiogenic aperture of the blood vessels required for the growing of new vascular branches, involving, among other processes, the loss of the BBB, transient edema, and destabilization of the basement membrane by the action of extracellular matrix proteases[22]. Therefore, the abnormal IB4 staining could be a deposit produced by the pathologic angiogenesis-mediated basement membrane mobilization.

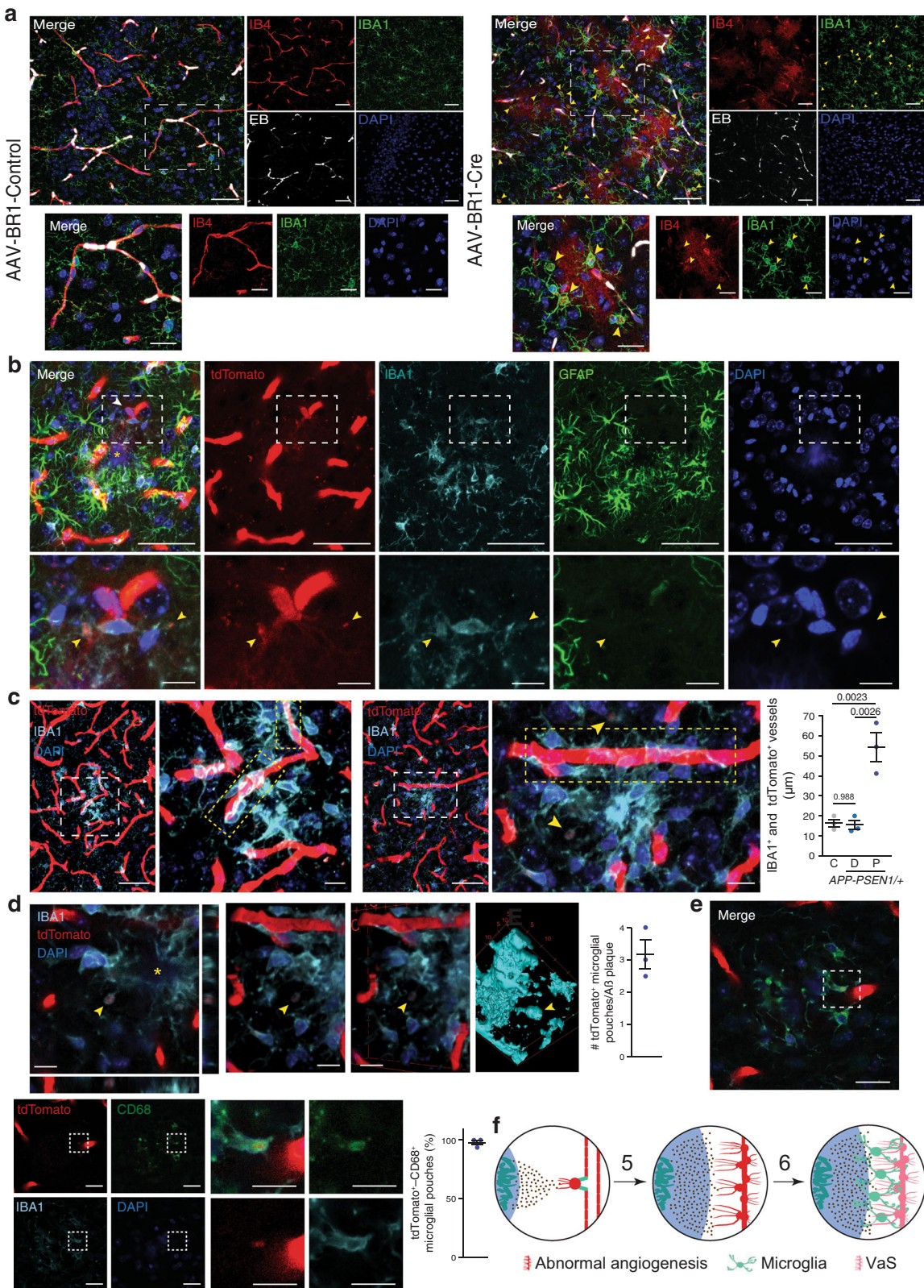

Interestingly, NPA induction (γ-secretase genetic inhibition) in adult endothelial cells is sufficient to produce identical IB4$^+$ VaS in the absence of amyloid pathology. In the human AD brain, (i) BBB disruption has been mapped to Aß-depositing areas that show brain parenchymal accumulation of blood-derived immunoglobulin G and Fibrin[17], (ii) microhemorrhages are a frequent characteristic of Aß plaques[48], (iii) Aß plaque-associated vascular ghosts (emanating from blood capillaries) and endothelial debris have been described using several endothelial basement membrane markers[49–53], and (iv) metalloproteases are induced[18], suggesting a similar defective angiogenic process. Under this arrested angiogenesis, VEGF may also act as permeabilizing factor, aggravating the phenotype and increasing the IB4$^+$ vascular deposit[54].

**Fig. 6 Microglial cells phagocyte VaS-associated blood vessels. a** Striatal confocal XY images from $Psen1^{loxP/loxP}$; $Psen2^{-/-}$ mice that were injected with cerebral endothelium-specific adeno-associated control (AAV-BR1-Control; C) or Cre recombinase-expressing (AAV-BR1-Cre; Cre) viruses, perfused with Evans blue (EB; white) and stained with endothelial (IB4; red), microglia (IBA1; green), nuclear (DAPI; blue) markers. Yellow arrowheads indicate microglial IB4+ pouches. Lower row images show the dashed white rectangles depicted in the upper row images. Scale bars = 20 and 10 μm in low and high magnification images. **b-e** Cortical confocal images 8-month-old $Cdh5$-$Cre::ERT2/+$; $R26$-$LSL$-$tdTomato/+$; $APP$-$PSEN1/+$ tamoxifen-treated mice and stained with a tdTomato antibody (**b**) or direct tdTomato fluorescence (**c-e**) (red) and with microglial (IBA1; cyan), astrocytic (GFAP; green —**b**), lysosomal (CD68; green —**e**) and nuclear (DAPI; blue) markers. Aß plaques are indicated with a yellow asterisk. Yellow arrowheads indicate internalization of tdTomato+ signal by microglial pouches. Lower row images show the dashed white rectangles depicted in the upper row (**b**, **e**) or left (**c**) images. A white arrowhead indicates a tip cell that projects extensions towards an Aß plaque (**b**). Scale bars (**b-e**) = 50 μm in low and 10 μm in high magnification images. Regions with high alignment of endothelial cells and microglia are highlighted with dashed yellow rectangles in **c**. Left panel in **d** shows a z-projection of a magnified and cropped image from **c** showing the orthogonal projections and the right panels show different rotated views of 3D reconstructions of the left image (volume, two central panels; surface, right panel). Right graphs (**c-e**) show the quantification of **c** the length of tdTomato+ vessels occupied by IBA1+ signal. Mean ± SEM. $n = 3$ mice; ANOVA, post hoc Tukey's test; **d** the number of microglial pouches proximal to Aß plaques. Mean ± SEM. $n = 3$ mice; and **e** the percentage of microglial IB4+/CD68+ pouches. Mean ± SEM. $n = 3$ mice. **f** Working model of the process of vascular scars (VaS) formation. Numeration continues from Fig. 1f.

Could NPA explain the reduced blood vessel density observed in AD? We postulate that reduction of the lateral inhibition in angiogenic endothelial cells could disassemble Aß plaque-associated blood vessels into non-perfused tip cells[7–10]. However, tip cells were not easily identified in the brain of an AD mouse model, suggesting that these cells with halted differentiation could be being removed. Using genetically labeled endothelial cells, we report that microglia strongly react to endothelial cells in the proximity of Aß plaques and engulf blood vessels producing phagocytic pouches[44]. This microglial reaction is similar to that observed in a mouse model of ischemic cortical stroke, where microglia were attracted by blood serum proteins released into the brain parenchyma and engaged in endothelial cell phagocytosis[55,56]. The disassembly of pre-exiting blood vessels around Aß plaques could be the consequence of microglial activation by Aß itself and/or the dysfunction caused by NPA in endothelial cells. However, γ-secretase genetic inhibition in adult endothelial cells induced a strong phagocytic phenotype in microglial cells in the absence of Aß accumulation, strongly suggesting that NPA induction is sufficient to elicit blood vessels phagocytosis by microglia. Interestingly, the molecular signature of reduced lateral inhibition contains, in addition to angiogenic and tip cell-enriched genes, several inflammatory mediators[34] that could induce the cross talk with AßAM. Other interesting players could be the astrocytes and the non-microglia brain resident macrophages. Astrocytes produce VEGF around Aß plaques, accumulate in the deposits, and may have a role in the degradation of the vessels, either by themselves or by modulating the phagocytic activity of the microglia. We cannot discard a possible role of other brain resident macrophages, as actually the only marker to discern between them and microglia is TMEM119 (ref. [57]), which is strongly downregulated in AßAM[58]. However, single-cell sequencing in an AD mouse model suggested a minor role of those cells compared with microglia[58] and a recent work reported that Aß plaque-associated myeloid cells derive from resident microglia[59].

Our results demonstrate that, at least in endothelial cells, inhibition of the γ-secretase could play a major role in AD pathology. fAD mutations in *PSEN1* and *PSEN2* reduce γ-secretase activity over NOTCH and APP, but increase the ratio between $Aß_{1-42}/Aß_{1-40}$ by reducing γ-cleavage following $Aß_{1-40}$[3,4]. Owing to the importance of the γ-secretase in the APP processing, a lot of effort has been made to understand the consequences of the neuronal loss of γ-secretase activity[60,61]. Interestingly, a reduction of only 50% in *Dll4* dose (heterozygous mice) is sufficient to induce NPA[7–10], demonstrating the exquisite sensitivity of the angiogenic system to subtle changes in the endothelial NOTCH pathway. Therefore, a vicious cycle can be perpetuating in AD between accumulation of $Aß_{1-42}$, an intermediate γ-secretase reaction product that could inhibit its activity[4], and reduction of blood local vessel function by NPA, which could decrease the local clearance and further stimulate Aß deposition.

AßAM have a key role in AD progression[62–64] and others and we have recently shown that AD microglia suffer from hypoxic and metabolic stress that compromise their protective activity[23,65]. Therefore, although Aß plaques might constitute an already late stage in the progression of AD, strategies to reestablish angiogenesis around Aß plaques, including reactivation of the γ-secretase/NOTCH pathway in endothelial cells, might hold therapeutic potential.

## Methods

**Mice**. Mice were housed under controlled temperature (22 °C) and humidity conditions in a 12 h light/dark cycle with ad libitum access to food and water. Housing and treatments were performed according to the animal care guidelines of European Community Council (86/60/EEC). Principles of laboratory animal care (NIH publication No. 86-23, revised 1985) were followed, as well as specific Spanish national laws where applicable. The competent Spanish authority approved all the procedures ("Consejería de agricultura, pesca y desarrollo rural. Dirección general de la producción agrícola y ganadera"). Mice showing any alterations at the moment of the allocation (wounds, smaller or bigger body size, etc.) were excluded. B6.Cg-Tg(APPswe,PSEN1ΔE9)85Dbo/J (*APP-PSEN1*; stock number 34832-JAX), *Cp-HIST1H2BB::Venus/+* (Tg(Cp-HIST1H2BB/Venus) 47Hadj/J; stock number 020942), and Ai14 Cre-reporter mice (B6.Cg-Gt(ROSA) 26Sortm14(CAG-tdTomato)Hze/J; stock number 007914) mice were obtained from Jackson Laboratories, *APP$_{751}$SL/+* mice[66] (Sanofis) were provided by Transgenic Alliance-IFFA-Credo, *Cdh5-Cre::ERT2/+*[67] was a generous gift from Prof. Ralf H. Adams and *Psen1$^{Floxed}$* and *Psen2$^{-/-}$* mice were kindly provided by Prof. Jie Shen. Only heterozygous *APP-PSEN1*, *APP$_{751}$SL*, Ai14, *Cp-HIST1H2BB:: Venus*, or *Cdh5-Cre::ERT2* mice were used. Experimental groups were homogeneously distributed by sex and assigned to each treatment without previous observation of the mice by the experimenter. No randomization methods were employed. Mice were euthanized by administration of a lethal dose of anesthesia (sodium thiopental, thiobarbital). To activate Cre::ERT2-mediated recombination, mice were fed for 30 days with a diet containing tamoxifen (400 mg tamoxifen citrate per kg; Envigo). Viral induced *Psen1$^{Floxed}$* Cre-mediated recombination in adult cerebral endothelial cells was achieved by AAV-BR1-Cre[40] injection in the tail vein ($5 \times 10^{10}$ genomic particles per mice). Angiography with Evans Blue was performed as described[35]. Sorafenib was injected at a dose of 30 mg/kg/2 day for 30 days. A summary of all the mice used in this article can be found in Supplementary Data 1.

**Human samples**. The use of brain tissue samples was coordinated by the local brain bank (Banco de Tejidos CIEN, Madrid, Spain), following national laws and international ethical and technical guidelines on the use of human samples for biomedical research purposes. In all cases, brain tissue donation, processing, and use for research followed published protocols, which include obtaining informed consent for brain tissue donation from living donors and the approval of the whole donation process by the Ethical Committee of the Banco de Tejidos CIEN (committee approval reference 15-20130110). Hippocampal samples included five samples classified with Braak tau pathology (Braak IV–VI) and five control (Braak 0–I) samples (Supplementary Data 1). For morphological studies, 4%

paraformaldehyde fixed samples (24 h) were sectioned (50 μm thickness) on a vibratome and serially collected in PBS and 0.02% sodium azide[33]. A summary of all the human samples used in this article can be found in Supplementary Data 1.

**EAE mouse model.** Seven-week-old female C57/BL6 mice were purchased from Charles River Laboratories (Wilmington, MA, USA) and acclimatized for a week in the appropriate environmental conditions at the Animal Facilities of the Instituto Cajal-CSIC prior to carrying out the procedure. To induce EAE, we follow the recommendations of the Spanish Network in Multiple Sclerosis-REEM and the procedure used in Dr. F. de Castro's group[68]. In brief: mice were anesthetized intraperitoneally with 40 μL of an anesthetic/analgesic mixture containing keta-mine and xylazine. An emulsion of Myelin Oligodendrocyte Glycoprotein (MOG$_{35-55}$ peptide, 250 μg in a final volume of 200 μL; GenScript) and complete Freund's adjuvant (CFA) containing inactivated *Mycobacterium tuberculosis* (4 mg; BD Biosciences) was induced subcutaneously into groin and armpits. Then, Per-tussis toxin (400 ng/mouse: Sigma-Aldrich) was intravenously administered in one of the lateral tail veins. This was also repeated 48 h later. Clinical score in animals was evaluated until sacrifice by two independent blind observers as follows: 0 = asymptomatic; 0.5 = paralysis of the distal tail; 1 = loss of muscle tone throughout the tail; 2 = weakness or unilateral partial hindlimb paralysis; 3 = bilateral paralysis of the hind limbs; 4 = tetraplegia; and 5 = death. Three experimental groups were organized: "onset" (clinical score ~0.5), "peak" (clinical score between 2.5 and 3), and "post-peak" (3 days after peak). Animals were sacrificed by intraperitoneal administration of a lethal dose of pentobarbital and they were perfused transcar-dially with 4% paraformaldehyde (PFA) in 0.1 M phosphate buffer (PB, pH 7.4). The encephala were obtained and post-fixed overnight in PFA 4% and glutar-aldehyde 0.025% at room temperature (RT). All procedures were performed in compliance with the ARRIVE Guidelines, in accordance with the Guidelines of the European Union (63/2010/EU, 90/219/EEC, Regulation No. 1946/2003) and fol-lowing the Spanish regulations (RD 53/2013, BOE 8/2/2013) for the use of laboratory animals. The generation of the EAE murine model of MS at the Instituto Cajal-CSIC has been properly approved by the institutional and regional ethics committees (references 2016/049/CEI3/20160411, CSIC440/2016 and PROEX143/16).

**Double ISH and immunohistochemistry (IHC).** Mice were euthanized and cer-ebral samples fixed as in ref. [35]. Tissues were cryoprotected in sucrose 30% in PBS at 4 °C for 24 h and embedded in OCT compound (Tissue-Tek) prior to −80 °C storage. Thirty-micrometer coronal slices were obtained with a cryostat (Leica) and stored at −80 °C until use. RNAscope 2.5 Brown (ACD) protocol was used to detect *Vegfa* (ACD probe 436961 Vegfa01), *Plaur* (ACD probe 48731 Plaur), or *Psen1* (ACD probe 451011 Psen1) mRNAs according to the manufacturer's instructions for frozen tissue, using a HybEZ oven (ACD). Subsequent immu-nostaining was performed for microglia (IBA1, 1:200), astrocytes (GFAP, 1:1:500), EC (IB4; 1:50), Thio-S, and nuclear staining (DAPI). After RNAscope 2.5 Brown protocol, slices were incubated for 10 min in PBS–0.3% Triton X-100 (v/v) and washed in PBS. Antibodies were prepared in PBS–0.05% Triton X-100, 2% normal goat serum (Gibco) and used to incubate the samples overnight at 4 °C. After several PBS washes, slices were incubated with anti-rabbit conjugated with Alexa-588 or 647 (Invitrogen, 1:400) for 1 h at room temperature. Several washes with PBS, Thio-S (0.005% in PBS; 8 min), and DAPI (Sigma, 1:1000; 5 min) were the final steps before mounting with Fluoromount-G.

**Immunodetection.** Human brain samples: a protocol adapted from ref. [35] to human brain samples was used. Briefly, brain sections were incubated in 50 mM NH$_4$Cl in PBS for 30 min, then in 50 mM glycine in Tris pH 8 for 5 min at 80 °C with gentle shaking, and finally in CaCl$_2$-containing buffer (0.1 mM CaCl$_2$; 0.1 mM MgCl$_2$; 0.1 mM MnCl$_2$ diluted in 0.1 M PBS pH 6.8) and heated for 90 s in a microwave (600 W). Sections were incubated for 72 h at 4 °C in blocking solution (CaCl$_2$-containing buffer, 0.05% (v/v) Triton X-100 and 2% (v/v) NGS in 0.1 M PBS) using primary antibody (anti-Iαvβ3, Abcam, 1:50), secondary antibody Alexa-568 anti-mouse (Molecular Probes, 1:500), and Thio-S post-staining to visualize Aβ plaques and quantify tangle density. Sections were then treated using the standard Eliminator (Merck Millipore) protocol and poststained with DAPI (1:1000). The images were then generated with the NewCAST system (Visiopharm) associated with the microscope BX61 (Olympus). Mouse samples: mice were anesthetized with an overdose of thiobarbital and perfused with an intracardial injection of Evans blue[35]. The brains were dissected and immediately fixed over-night at 4 °C with the fixation solution (4% paraformaldehyde in PBS–0.05% glutaraldehyde). The brains were cryoprotected during 48 h with a solution of 30% sucrose in PBS and embedded in OCT. Blocks were sliced in 40-μm-thick coronal sections using a cryostat (CM 1950, Leica). Tissues showing evident technical alterations (i.e. not properly fixed, stained or cut) were excluded. Immunostaining was performed on free-floating sections according to the Wälchli et al.[35] protocol. Brain sections were post-fixed in fixation solution and, for antigen unmasking, sections were incubated in 50 mM NH$_4$Cl in PBS for 30 min, then in 50 mM glycine in Tris pH 8 for 5 min with gentle shaking at 80 °C, and finally in CaCl$_2$-containing buffer (0.1 mM CaCl$_2$; 0.1 mM MgCl$_2$; 0.1 mM MnCl$_2$ diluted in 0.1 M

PBS pH 6.8) and heated for 90 s in a microwave (600 W). Sections were incubated for 72 h at 4 °C in blocking solution (CaCl$_2$-containing buffer, 0.05% (v/v) Triton X-100, and 2% (v/v) NGS in 0.1 M PBS) using primary antibodies (anti-IBA1, Wako, 1:400; anti-GFAP, Sigma, 1:1000; anti-AQP4 1:5000; anti-TER119, Invi-trogen, 1:400; anti-mCherry, EnCor Biotechnology, 1:1000—to visualize tdTomato; and anti-CD68, Bio-Rad, 1:100) or biotinylated IB4 lectin (Sigma, 1:50). For immunofluorescent studies, we used secondary antibodies anti-mouse or anti-rabbit conjugated with Alexa-488, Alexa-568, or Alexa-647 (Molecular Probes, 1:800), and streptavidin conjugated with Alexa-488, Alexa-568, and Alexa-647 (Jackson, 1:500). Microwave heating was excluded when anti-Laminin (Sigma, 1:250) and PDGFR-ß (Invitrogen, 1:200), antibodies were employed. Anti-CD31 (BD Biosciences, 1:500) staining was performed in unfixed sections for 1 h, fixed for 15 min, and the IHC performed as described before. Thio-S (Sigma, 0.005% in PBS) and DAPI (Sigma, 1:1000) were used as counterstains according to the standard procedures. Hypoxia staining was performed in three 14-month-old *APP-PSEN1* mice injected intraperitoneally with 60 mg/kg of Pimonidazole HCl and sacrificed 45 min after injection. Brain was snap frozen in liquid nitrogen, sectioned in a cryostat (20 μm, Leica), and sections were fixed for 10 min in cold acetone. Protein–pimomidazole hypoxic adducts were detected using a polyclonal primary antibody (Hypoxiprobe, 1:50, PAb2627AP) and Aβ plaques were counter stained with Thio-S (green). WT mice injected with pimonidazole HCl and *APP-PSEN1* mice without injection were used as negative controls and kidney was used as a positive control.

**Imaging.** Unless otherwise stated, all fluorescent images of cortical regions of brain sections from mouse brains were acquired in a confocal microscope (Nikon A1R+) in Z-stack series and colocalization images with DAB were performed by decreasing opacity of the fluorescent images.

**Electron microscopy.** For EM-gold labeling, 50 μm vibratome sections from *APP-PSEN1/+* mice hippocampus (fixed with 4% paraformaldehyde/75 mM lysine/10 mM sodium metaperiodate) were cryoprotected in a 25% sucrose and 10% glycerol solution and then frozen at −80 °C in order to increase IB4-lectin-binding effi-ciency. Sections were incubated in biotinylated IB4 (Sigma, 1:100), followed by 1.4 nm gold-conjugated streptavidin (Nanoprobes, 1:100). The tissue was then post-fixed in 2% glutaraldehyde and washed in 50 mM sodium citrate. HQ Silver Enhancement Kit (Nanoprobes) was used and gold-toning was performed. Sections were then fixed in osmium tetroxide, block-stained with uranyl acetate, dehydrated in graded acetone, and flat-embedded in Araldite (EMS). Finally, sections were cut in ultrathin sections (70 nm) and examined under a transmission electron microscope (JEOL JEM 1400).

**Image quantification.** Human Integrin/Aβ plaques study: we have developed a method to quantify, measure, and compare the location of integrins and Aβ pla-ques in two-dimensional images of hippocampus biopsies. This approach consisted of two steps applied to every biological sample: first, we measured Aβ plaques over the natural biopsies, and second, we fixed the Aβ plaques positions while rando-mized the integrins locations over the region of interest (ROI). The ROIs were defined by the biological sample contained on the image, excluding artefacts and empty regions. (i) Biopsy measurements: for each integrin marker we have created a geodesic distance image using the ROI, in which this marker position defined the origin of coordinates. Then, we captured the chessboard distance assigned to every pixel position matching with the Aβ plaques markers locations. Thus, we measured the distance from each integrin marker to the closest Aβ plaque and computed its average. (ii) Randomizing integrin markers positions: we carried out a randomi-zation protocol repeated 500 times for each biological sample. We fixed the integrins markers positions and randomized the Aβ plaques ones along the ROI, thus we measured the distances between the markers as described above. (iii) Finally, we compared the minimum distances between integrins and the Aβ pla-ques obtained in each raw sample and its corresponding 500 randomizations. For statistical analysis, only ten randomized simulations were used. Area+/cell number density of different markers: all the measurements were performed in the cortices of 8-month-old *APP-PSEN1/+*, *APP$_{751}$SL/+*, or WT mice, cortices, striata, and hippocampi of *PSEN1$^{Flox/Flox}$; PSEN2$^{-/-}$* mice. Cortical XY confocal twin images (between 5 and 10 images per mice) containing centered Aβ plaques (rigorously scrutinized by Thio-S labeling or blue autofluorescence, when required) and adjacent brain regions without plaques were used. A circumference of 100 μm of diameter was drawn in the center of each Aβ plaque imaged and the area occupied by the VaS was also drawn and quantified. Laminin+ areas and *Plaur* and *Psen1* mRNA cell+ number were manually outlined/counted and quantifications were performed using Fiji (v. 2.0.0). Other markers were measured using a R-based semi-automatic process to hide the name of the samples and obtain the area occupied by every marker using Fiji. Quantification of VaS load in AD mouse models: all the measurements were performed in total cortical area of WT (14-month-old), *APP-PSEN1/+* (8- and 12-month-old), and *APP$_{751}$SL/+* (8-month-old) mice. Quantifications were done in superimages generated with the NewCAST system (Visiopharm) associated with the microscope BX61 (Olympus). VaS load was measured using Fiji. A segmented binary mask was generated and the occupied

area by detected particles over a specific constant threshold was quantified. Load was defined as the percentage of total cortical area occupied by VaS. Quantification of VaS and Plaque areas: all the measurements were performed in total cortical area of 8- or 12-month-old *APP-PSEN1/+* or *APP₇₅₁SL/+* mice. Quantifications were done in images generated with the upright BX61 microscope (Olympus; ×40 objective). VaS and Thio-S areas were measured using Fiji. A segmented binary mask was generated and the occupied area by detected particles over a specific constant threshold was quantified.

**Flow cytometry**. Mice were anesthetized and transcardially perfused with HBSS (−CaCl₂/−MgCl₂) (Gibco) and cortices were dissected and then dissociated using a Tissue Chopper (Vibratome, 800 series). Chemical digestion was then performed with a mix of papain (Worthington) (8 U/mL) and DNAse I (Sigma; 80 Kunitz units/mL) followed by a Percoll gradient (GE Healthcare) at 90% in PBS (v/v). Cells were stained with primary conjugated monoclonal antibodies CD11b-APC (eBioscience) and CD31-PE (BD Bioscience) diluted 1:200 at 4 °C for 30 min. Staining with isotype control-PE and isotype control-APC (eBioscience, 1:200) was used as a negative control. Both control and experimental samples were simultaneously incubated with anti-CD16/CD32 blocker (eBioscience, 1:200). Cells were washed and sorted using a FACS Aria Fusion (Becton Dickinson) flow cytometer and data acquired and analyzed with FACSDiva software 8.0 (Becton Dickinson). Gating strategy and data analysis were done according to the guidelines[69]. Debris and dead cells were discarded by forward and side scatter pattern. FSC-A and FSC-H events distribution was used to gate single cells (Supplementary Fig. 2). Endothelial cells were identified as positive events for CD31 and negative for CD11b. Percentages are relative to total single cells.

**DNA and RNA extraction and quantitative reverse transcriptase (qRT)-PCR and qPCR**
*FACS-isolated endothelial cells and mouse brain samples*. RNA was extracted from FACS-isolated endothelial cells using TRIzol reagent (Life Technologies) and DNA from mouse brain areas using DirectPCR (Viagen). RNA samples (full RNA extracted from isolated cells) were treated with PerfeCTa DNase (Quanta Biosciences) and copied to cDNA using qScript cDNA Supermix (Quanta Biosciences). Real-time q(RT)-PCR was performed for all samples in a ViiA 7 Real-Time PCR System (Applied-Biosystems) using Power SYBR-Green PCR Master Mix (Applied-Biosyor iTaq Universal Probes Supermix, Bio-Rad) (Primers: *Cdh5*: 5′-TCTCTGCAACAGACAAGGATGTG-3′, 5′-TGTTGGCGGTGTTGTCATG-3′; *Cd33*: 5′-GAGGCAGGAAGCGATCACAT-3′; 5′-GTGTATGGAACATCCTGG AGTCAC-3′; *Gfap*: 5′-GCCACCAGTAACATGCAAGAGA-3′; 5′-TGCAAACTT AGACCGATACCACTC-3′; *Gstp1*: 5′-ATGCCACCATACACCATTGTC-3′; 5′-G GGAGCTGCCCATACAGAC-3′; *Hmbs*: 5′-CCATACTACCTCCTGGCTTTACT ATTG-3′; 5′-GGTTTTCCCGTTTGCAGATG-3′; *Iba1*: 5′-ATCAACAAGCAAT TCCTCGATGA-3′; 5′-CAGCATTCGCTTCAAGGACATA-3′; *Psen1*: 5′-TGAGC CAATTCAAGCCAGAGT-3′; 5′-TGTGTGTGGTCTGTGAAGAGT-3′; *Stt*: 5′-AC CGGGAAACAGGAACTGG-3′; 5′-TTGCTGGGTTCGAGTTGGC-3′; *Stnb2*: 5′-A ACACCTTGATCTTACGCTGCAA-3′; 5′-GCCTCCCGCTGTACTGGTT-3′).

**Microarray**. The RNA quality was analyzed using an Agilent 2100 Bioanalyzer (Agilent). Only samples with RNA integrity number (RIN) higher than 7 were further processed for microarray analysis. RNA was amplified and labeled using the GeneChip WT Pico Reagent Kit (the total RNA isolated was used as the starting material; Affymetrix). The amplified cDNA was quantified, fragmented, and labeled in preparation for hybridization to GeneChip® Mouse Transcriptome 1.0 Array (Affymetrix) using 5.5 μg of single-stranded cDNA product and following protocols outlined in the user manual. Washing, staining (GeneChip® Fluidics Station 450, Affymetrix), and scanning (GeneChip® Scanner 3000, Affymetrix) were performed following the protocols outlined in the user manual for cartridge arrays. Raw data from the extraction software Expression Console (Affymetrix) were imported to the R statistical processing environment (RStudio, Inc.) using the LIMMA/Bioconductor package. Quality of the data was assessed using Array Quality Metrics package. Data were normalized using the Robust Multi-Array (RMA) method and differential expression analysis was done using LIMMA/Bioconductor package. The data discussed in this publication have been deposited in NCBI's Gene Expression Omnibus and are accessible through GEO Series accession number GSE97423. A list of the DE genes in the retina of *Dll4+/−* mice[34] was obtained from their authors. To identify underlying biological processes in endothelial cells from *APP-PSEN1/+* mouse models, we used GSEA. We analyzed the enrichment of 825 gene sets from the Biological Processes Database C5 version 5.1 and the custom GS Dll4+/−_Up (50 most upregulated genes—FDR *p* < 0.05—in ref. [34]).

**iDISCO**. Brains from WT and *APP-PSEN1/+* mice were clarified following the iDISCO protocol[70] with some modifications. Brains were consecutively perfused with PBS-Evans blue and the fixation solution (4% paraformaldehyde in PBS–0.05% glutaraldehyde). Samples were not treated with methanol to preserve IB4 staining, blocked for 3 days at 37 °C, and both the wash solution and the primary antibody solution were prepared using CaCl₂-containing buffer, to improve the filopodia and endothelial cell staining. The brains were incubated in

IB4 (Sigma, 1:50) for 3 days at 37 °C and a Cy3 conjugated streptavidin (Jackson, 1:500) was used for 3 days at 37 °C. In all incubation steps azide 0.02% was included to avoid contamination. For nuclear labeling, immunolabeled samples were incubated with DAPI (1:1000 in PBS–0.2% Tween–20; 10 μg/mL heparin). For brain clarification, samples were incubated overnight in 10 mL of 50% v/v tetrahydrofuran–H₂O (THF; Sigma), 10 mL of 80% THF–H₂O for 1 h, twice in 100% THF for 1 h, and then in dichloromethane (Sigma) until samples sank at the bottom of the vial. Finally, samples were incubated in 18 mL of dibenzyl ether (DBE, Sigma) until clear (2 h) and then stored in DBE at room temperature.

**iDISCO imaging and image processing**. Three-dimensional (3D) imaging of brain samples was performed with a bidirectional light sheet microscope (Ultramicroscope II, LaVision BioTec), controlled by the InspectorPro Software (LaVision BioTec), with a stereomicroscope (MVX10, Olympus, Japan) equipped with a ×2 objective (MVPLAPO, Olympus). Images were acquired using a Neo sCMOS camera (Andor, Oxford Instruments) at a magnification of ×0.8, ×1.6, and ×6.3, with the tissue submerged in Ethyl Cinnamate. Cy3 was excited with laser line 561 nm and collected with a BP 620/60 filter. To avoid saturated pixels, laser intensities were set at 4–6%. The numerical aperture of the light sheet was set at 0.078, so the step size was fixed at 2 μm. Imaris software (Bitplane, http://www.bitplane.com/imaris/imaris) was used to generate 3D volume and movie files, using the volume rendering function and the snapshot and animation tools.

**Statistics and reproducibility**. All individual measurements constitute independent biological replicates and the experiments were repeated at least three times (for quantifications, the *n* is specified in the figure legends). Samples with an *n* < 9 were analyzed using parametric tests. Samples with an *n* ≥ 9 were evaluated for normal distribution using the D'Agostino and Pearson's omnibus normality test. Comparisons between two groups were performed with two-sided unpaired Student's *t*-test, whereas comparisons between more than two groups were done with ANOVA with Tukey's test. Data not adjusting to a normal distribution were analyzed using non-parametric test (Kruskal–Wallis test followed by Dunn's multiple comparison test). Data expressed as mean ± SEM; *p* ≤ 0.05 were considered statistically significant. Statistical analyses and graphs were performed in GraphPad Prism version 6.0 (GraphPad Inc.).

**Reporting summary**. Further information on research design is available in the Nature Research Reporting Summary linked to this article.

## Data availability
The raw data of endothelial cells microarray are available at the Gene Expression Omnibus under accession code GSE121729. Source data are provided with this paper.

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

## Acknowledgements

A.E.R.-N. was the recipient of a JdlC-F fellowship from the Spanish Ministry of Economy, Industry, and Competitiveness (MINEICO) (FJCI-2015-23708), M.I.A.-V., N.L.-U., and C.O.-d.S.L. were the recipient of an FPU fellowship from Spanish Ministry of Education, Culture, and Sport (respectively, FPU15/02898, FPU14-02115, and AP2010-1598), and R.M.-D. was the recipient of a "Sara Borrell" fellowship from ISCIII (CD09/

0007). Work was supported by grants to A.P. by the Spanish MINEICO, ISCIII, and FEDER (SAF2012-33816, SAF2015-64111-R, RTI2018-096629-B-100, SAF2017-90794-REDT, and PIE13/0004), by the regional Government of Andalusia ("Proyectos de Excelencia", P12-CTS-2138 and P12-CTS-2232) co-funded by CEC and FEDER funds, and by the "Ayuda de Biomedicina 2018", Fundación Domingo Martínez; J.Vitorica: Instituto de Salud Carlos III (ISCiii) of Spain, co-financed by FEDER funds from European Union (PI18/01556) by La Marató-TV3 Foundation grant 20141431; by CIBERNED (CB06/05/0094); and by Junta de Andalucia Consejería de Economía y Conocimiento through grant US-1262734; A.G.: Instituto de Salud Carlos III (ISCiii) of Spain, co-financed by FEDER funds from European Union, through grant PI18/01557; and by Junta de Andalucia Consejería de Economía y Conocimiento through grants UMA18-FEDERJA-211 and P18-RT-2233 co-financed by Programa Operativo FEDER 2014-2020. The authors thank Maria Llorens-Martin (CBM-Severo Ochoa, Madrid, Spain) for the generous gift of the human samples; Ralf H. Adams and Jose L. de la Pompa for providing the Cdh5-Cre::ERT2 and *Cp-HIST1H2BB::Venus* mice; K. Levitsky (microscopy), M.J. Castro (flow cytometry), F.J. Moron (genomics), and R. Duran (histology) for advice and technical assistance in experiments at IBiS core facilities and to Dr. Alberto Serrano-Pozo for his critical review of the manuscript. We thank Sanofi (France) for the $APP_{751}SL$ model used in this work.

## Author contributions

These authors contributed equally and are listed by alphabetic order: M.I.A.-V. and A.E.R.-N., A.P., M.I.A.-V., and A.E.R.-N.: conceived and designed research. A.P., M.I.A.-V., A.E.R.-N., R.M.-D., G.R.-P., N.L.-U., C.O.-d.S.L., M.A.S.-G., J.L.T.-C., M.M.-B., A.S.B.-B., J.C.D., and R.G.-M. performed research. A.P., M.I.A.-V., A.E.R.-N., G.R.-P., M.M.-B., A.S.B.-B., P.G.-G., P.V.-M., and L.M.E. analyzed the data. L.M.E., J. Villadiego, M.E., F.d.C., A.G., J. Vitorica, A.R., E.H., R.d.T. provided methodological and/or scientific assistance. A.R. contributed with the human brain specimens. J.V., F.G.S., A.C.S.-H., B.F.-G., M.A.M., and F.d.C. contributed with mouse models/samples. M.T. and J.K. contributed with AAV vectors. M.I.A.-V. and J.K. critically reviewed the manuscript. A.P. and A.E.R.-N. wrote the manuscript.

## Competing interests

The authors declare no competing interests.
