## [Peer Review File · Nature Communications]

Reviewers' comments:

Reviewer #1 (Remarks to the Author):

This paper investigates the disrupted angiogenesis surrounding amyloid plaques in a model of brain amyloid accumulation related to Alzheimer's disease. Molecular markers of non-productive angiogenesis (impaired lateral inhibition) were observed around amyloid plaques in APP-PS1 mice and AD brains. Microarray analysis of sorted endothelial cells implicated a gene set also observed in a mouse model of non-productive angiogenesis caused by deficiency of the notch ligand Dll4. In addition to blood vessels, isolectin B4 immunostaining also identified positive areas near the plaques in APP mice, a pattern observed also with endothelial deletion of presenilin-1 in WT mice using AAV-BR1. Administration of a tyrosine kinase inhibitor reduced plaque load in APP-PS1 mice, an effect attributed to increased clearance of Abeta due to rescue of the angiogenesis disruption. It is concluded that non-productive angiogenesis is a failed compensatory mechanism that promotes amyloid pathology by impairing Abeta clearance.

This study raises a number of questions concerning the mechanism of the microvascular remodeling and its impact on the disease process:

It is assumed that the triggering factor is hypoxia (fig. 1). Yet, no evidence is presented that there is hypoxia in the affected areas, which, in turn, leads to disturbed angiogenesis. First hypoxia needs to be documented and then the causes would need to be defined and the downstream signaling identified.

Assuming that hypoxia is the triggering event, VEGF mRNA seems to be expressed more in Iba1+ cells than astrocytes (Fig. 1B,C), but the quantification shows more positive astrocytes (Fig 1D). Along the same lines, more robust methods are needed to document and quantify the expression of the integrin alpha-v-beta3 (Fig. 1 E,F).

The significance of the Evans blue (EB) experiments to assess microvascular perfusion is also unclear. The assumption is that the lack of EB fluorescence reflects non-patent vessels, but global cerebral perfusion is reduced in these mice and there are also microvascular occlusions by circulating leukocytes (Nat. Med. 21, 880; 2015). More quantitative methods to assess microvascular perfusion around the plaques and elsewhere would be needed.

In the experiments with AAV-BR1 in WT mice, the efficacy in gamma secretase inhibition and the cerebral endothelial selectivity of the effect needs to be demonstrated. It is unclear what the reduction in the pattern of EB stain and patchy IB4 stain mean.

It is also unclear how these results targeting PS1 deletion to endothelial cells relate to those in APP-PS1 mice in which the PS1 mutation is directed preferentially to brain cells with the prion protein promoter.

Concerning the role of microglia, Iba1 not a specific microglial marker and a role of brain macrophages cannot be ruled out. The reported phagocytic activity may be related to tissue remodeling caused by inflammation around the plaques, rather than specifically targeting the vessels.

The results of the experiment using ICP as a surrogate for ISF bulk flow/CSF clearance are difficult to interpret considering the wide variety of structural and functional factors that can alter ICP.

Similarly, the data with sorafenib are difficult to interpret due to wide variety of pharmacological effects of the drug (c-raf, NFkb, etc.). For example, it has been previously shown that the drug is protective in APP mice, an effect attributed to suppression of inflammation (e.g, Neuroscience. 2009;162:1220).

Finally, the cognitive impact of the treatment needs to be assessed and the ratio between Abeta40 and 42 determined.

Reviewer #2 (Remarks to the Author):

The authors report that A-beta deposits are surrounded by abnormal vascular structures. These structures contain VEGF but angiogenesis is impaired are there are some signs of vasoregression. The study addresses a very interesting finding in Alzheimer brains which others have reported before: vascular alterations around A-beta deposits. This study now aims to explore the underlying mechanisms. The authors present many interesting and provoking findings. The overall concept is pretty intriguing. Nevertheless the data do not (yet) justify the multiple conclusions drawn by the authors. The study is still preliminary in many aspects.

Major points:

I agree that the structures and the gene signatures are somehow reminiscent of Dll4/Notch1 or Rbpj deficient mouse models. However, to link the pathology to endothelial Notch signaling alterations a much deeper analysis would be required. Why do the authors use a PS1/2 gene deletion model instead of a Dll4/Notch model. Presenilins do not only cleave Notch receptors but have several additional substrates.

Also the description of the AAV mediated PS1/2 deletion in the brain vasculature appears superficial. Most importantly, it remains unclear how specific this procedure is for targeting the genes only in endothelial cells. Moreover, does this cause increased permeability, BBB breakdown, edema, etc. It is amazing to see that the quiescent brain vasculature can be altered in such a dramatic manner. It would be important to analyze how fast the vascular changes occur. Also, what is the percentage of endothelial cells targeted by the AAV and how many of these cells are involved in the formation of vascular lesions. Do the endothelial cells in such lesion proliferate? Are there signs of endothelial-to-mesenchymal transformation? Is there a dropout of pericytes only at lesions?

The authors have crossed mice with the Cdh6-Cre-ERT2. Why don't they use this cre to delete PS1/2 in blood vessels in adult mice instead of the AAV approach?

The use of Sorafenib gives interesting results. However, this RTK inhibitor has numerous substrates and is not very specific for inhibiting angiogenesis. The authors should use a VEGF inhibiting approach what would also fit better to the findings about increased VEGF levels. It also needs to be noted that VEGF is not only an angiogenesis stimulus but also a permeability factor. Altering permeability alone could also explain some of the findings.

If it is really dysregulated Notch signaling in endothelial cells that explains the vascular alterations, why do the authors not attempt to rescue it with a Notch gain of function approach.

There is absolutely no explanation in this paper why A-beta could interfere with Delta Notch signaling in the endothelium.

Reviewer #3 (Remarks to the Author):

Rosales-Nieves & Alvarez-Vergara et al. perform experiments suggesting that there are more angiogenic markers near amyloid-beta plaques in APP transgenic mice that are potentially derived from astrocytes. Floxing out PS1 from endothelial cells appears to increase an endothelial marker IB4 that may be not-productive. The findings are interesting and contribute to a new idea related to how Abeta may impact the brain by potentially inducing non-productive/functional vascular components. However, without functional data, it is a bit difficult to determine whether this marker does indeed represent functionless vessels or pathological angiogenesis. There also appears to be Iba1+ microglia that the authors are calling "vascular scars" near the amyloid-beta plaques that are engulfing vascular material. Interestingly, sorafenib (a kinase inhibitor) that decreases angiogenesis decreases amyloid-beta plaque load. They suggest that this may be due to enhanced Abeta clearance from the parenchyma though there is no data to directly support this mechanism.

The authors also make other conclusions in this manuscript, but many experiments with appropriate controls are suggested to reach those conclusions. Overall, Overall, the findings here have an opportunity to make new contributions to the literature but a number of issues need to be addressed.

Major points:

1) Most of the statistics in this paper are not performed in the appropriate fashion. Each plaque or cell is counted as an individual biological replicate, when instead each mouse or human should represent a single sample. For instance, Fig 1E suggests that there are n=2995 human brain samples, but there are in fact only n=4 samples. These analyses inflate the sample size and bias statistics towards being significant. The authors need to re-calculate their statistics based on individual samples, and they should NOT be representing each cell/image/brain section as an individual biological replica. Furthermore, all images should be quantified and analyzed with appropriate statistics.

2) Fig 1:

a. Fig 1D-G: The methods for the quantification need to be described. For example, in Fig 1C, were non-microglial cells assumed to be astrocytes? How many regions were imaged per mouse? Which regions? Etc. How was cell density calculated in Fig 1E?

b. Fig 1F: Given that there are human brains available, the authors should additionally reproduce in their hands whether there are more Integrin α v β 3+ cells in humans with AD vs. controls, in addition to the mouse data where they show that there are more Integrin α v β 3+ cells in APP/PS1 mice vs. controls.

c. If the authors want to conclude that angiogenesis is concentrated around plaques, a necessary experiment is to compare their data to how much angiogenesis exists outside of plaques. They do this in Fig 4A where there is a distal vs proximal plaque comparison; they should perform a similar analysis for this figure. Without this comparison, there might appear to be more angiogenesis around plaques when there is actually just an upregulation of angiogenesis everywhere in the brain. Furthermore, with the current set of experiments, the authors cannot conclude that pathological angiogenesis is "initiated" around plaques. This would require experiments with mice taken multiple timepoints starting before plaque onset.

3) Fig 2:

a. The age of APPPS1 mice vary a lot here (8, 10, 12, 18 mo). Please justify the different ages and why a single age was not used. Some phenotypes seen may be age-dependent.

b. Staining for amyloid-beta plaques in some way (rather than relying on their autofluorescence) is crucial for their identification.

c. Fig 2D: The authors claim that there is reduced reperfusion in 12 mo APP mice compared to 8 mo APP mice, however a quantification is needed for the Evans Blue staining, and this quantification needs to be compared to a WT control to make this claim.

d. Fig 3G also needs comparisons against a WT treated mouse.

e. It would be worthwhile to show that the lack of Evans Blue staining correlates with stronger IB4 staining, which would suggest that indeed these vessels are less productive.

f. It is surprising that in the microarrays, there was no change in IB4 or VEGF genes. To corroborate the claim that angiogenesis is non-productive around plaques, staining or in situ hybridization of microarray genes (Fig 2B) around plaques would support this claim.

4) Fig 3:

a. Conditional knockout of PS1 in endothelial cells using AAV-BR1-cre shows an increase in IB4 coverage, and this needs to be quantified compared to AAV-controls.

5) Fig 4 and Fig S3:

a. The data in Fig 4A is convincing in showing that there is laminin loss proximal to plaques. A great control would be to show the expected level of laminin in a WT mouse with no pathology in just the distal regions.

b. To implicate that there are less PDGFR β + pericytes, CD31+ endothelial, and astrocytic end-feet, quantification is needed (Fig S3A, B, C; Fig 4B).

6) Fig 5 and Fig. S4:

- a. Both figures need quantification. How many Iba1+ microglia are there around blood vessels distal or proximal from plaques? Which regions were imaged?
- b. If the authors claim that the tdTomato+ microglial pouches are phagocytic microglia, then a CD68 stain (or other marker for phagocytic process) for phagocytic microglia should be performed. The authors should be able to see vascular components inside the lysosomes of microglia.
- c. To show that microglia (and not other cell types like astrocytes) are surrounding blood vessels to create vascular scars, a comparison between the number of microglia or astrocytes surrounding blood vessels is necessary.
- d. The thickening of microglial projections also needs to be quantified.
- e. It would be worthwhile to discuss the relationship between astrocyte-derived VEGF and its relationship with microglia surrounding vessels and colocalizing with IB4.

7) Fig 6:

- a. It is very interesting that there is less intracranial pressure in APP transgenic mice compared to WT mice, potentially indicating less ISF bulk flow. To prove that sorafenib does indeed improve clearance and bulk flow, the authors should show that ISF bulk flow is restored with ICP.
- b. IB4 was used as a proxy for angiogenic endothelial cells. To confirm that more endothelial cell expression is indicative of angiogenesis, a classic marker for angiogenesis should also be used, such as Tie2.

Minor points:

- 1) State the sex of the mice for all experiments.
- 2) Fig 2: Clarify why Dll4+/- mice were used (i.e. angiogenesis is down regulated in these mice). Introduce APPSL+ mice and justify why these mice were used in addition to the APPPS1 mice.
- 3) State frequency/duration of sorafenib treatment.

Point-by-point response to the reviewers' commentaries

(Reviewers criticisms have been numerated to facilitate the reading. Original comments are in black and our responses in green. References to new text or figures are showed in red).

Reviewers' comments:

Reviewer #1 (Remarks to the Author):

This paper investigates the disrupted angiogenesis surrounding amyloid plaques in a model of brain amyloid accumulation related to Alzheimer's disease. Molecular markers of non-productive angiogenesis (impaired lateral inhibition) were observed around amyloid plaques in APP-PS1 mice and AD brains. Microarray analysis of sorted endothelial cells implicated a gene set also observed in a mouse model of non-productive angiogenesis caused by deficiency of the notch ligand Dll4. In addition to blood vessels, isolectin B4 immunostaining also identified positive areas near the plaques in APP mice, a pattern observed also with endothelial deletion of presenilin-1 in WT mice using AAV-BR1. Administration of a tyrosine kinase inhibitor reduced plaque load in APP-PS1 mice, an effect attributed to increased clearance of Abeta due to rescue of the angiogenesis disruption. It is concluded that non-productive angiogenesis is a failed compensatory mechanism that promotes amyloid pathology by impairing Abeta clearance. This study raises a number of questions concerning the mechanism of the microvascular remodeling and its impact on the disease process:

We thank the reviewer for the time employed in the evaluation and for his/her commentaries that have, without doubt, improved the quality and readability of our manuscript.

1.1. It is assumed that the triggering factor is hypoxia (fig. 1). Yet, no evidence is presented that there is hypoxia in the affected areas, which, in turn, leads to disturbed angiogenesis. First hypoxia needs to be documented and then the causes would need to be defined and the downstream signaling identified.

We apologize for our assumptions. We have now added new experimental work using pimonidazole hydrochloride (hypoxyprobe-1), which is able to label extremely hypoxic regions. These hypoxic regions can be detected by conventional immunohistochemistry using an antibody against the adducts generated between the probe and proteins in the hypoxic areas. We show that A β plaques are positive for the hypoxia marker (**Fig. 1B**).

The identification of the events leading to local hypoxia is the main focus of our manuscript. We show the regression of vessels and the lack of proper perfusion and endothelial cells around A β plaques (see below), therefore, at a final point, A β plaque-surrounding parenchyma is hypoxic. The initial events are far more complicated, as we do not have a way to know the age of an individual A β plaque. *In vivo* studies have shown that A β plaques grow for few days to its final size¹, so it is almost impossible to date plaques to distinguish between initial and final events. We propose that the displacement of the even distribution of blood vessels by the extracellular accumulation of a proteinaceous deposit (A β) can induce local and mild initial hypoxia (**Page 6, lines 3 to 5**).

Regarding the downstream signaling, hypoxia induces in the nearby cells the expression of VEGF (**Fig. 1C–E** and **Fig. S1A**), the accumulation of angiogenic $\alpha v\beta 3^+$ cells (**Fig. 1F–G**, **Fig. 2**, and **Fig. S1B–D**), the failure in the lateral inhibition (**Fig. 3**, and **Figs. S2, S3**), and the regression of the blood vessels by the activity of microglia (**Figs. 5, 6** and **Figs. S5, S7**), which perpetuates and extends the initial hypoxia. It is worth mentioning that endothelial cells, the main focus of this manuscript, do not experience hypoxia, but they are activated by the signaling emanating from hypoxic cells. We agree with the reviewer that the study of the hypoxia signaling in other cells is of paramount importance, and we have addressed this in a different manuscript, which we have now made available for the reviewers as complementary information to this review (see **March-Diaz et al.**). We show that microglia, the cell type found closest to A β plaques, express high levels of the hypoxia inducible factor 1 α (HIF1 α) and induces a HIF1-mediated transcriptional program (**Reviewers' Fig. 1 and 2**). Interestingly, this HIF1-mediated program is not observed in microglia from other neurodegenerative mouse models (**Reviewers' Fig. 1 and 2**). These findings have been published in a repository, have now been cited in our article², and are actually under second review in Nat. Aging.

1.2. Assuming that hypoxia is the triggering event, VEGF mRNA seems to be expressed more in Iba1+ cells than astrocytes (Fig. 1B,C), but the quantification shows more positive astrocytes (Fig 1D).

The figure shows the quantification of the number of VEGF⁺ cells that are also IBA1⁺ or GFAP⁺. We did not quantify the intensity of the signal. Indeed, we found very few A β plaques-associated microglia expressing VEGF (0 in around 100 microglial cells from two mice and 6 in around 150 microglial cells from three mice). These results are in perfect agreement with the low and high basal expression levels reported for *Vegfa* in microglia and astrocytes respectively^{3,4} (see **Reviewers' Fig. 3**). To avoid any potential confusion, we have now included "Number of" in the title of the Y-axis in **Fig. 1E** in addition to the already existing description of the Y-axis in the figure legend.

1.3. Along the same lines, more robust methods are needed to document and quantify the expression of the integrin alpha-v-beta3 (Fig. 1 E,F).

We have now improved our quantifications in mouse (Fig. 1G) and in human samples (Fig. 2 and Fig. S1B–D). Regarding mouse samples, we have quantified the density of $\text{I}\alpha\text{v}\beta\text{3}^+$ cells in wild-type mice and in AD mouse models both distal and proximal to A β plaques. The quantification clearly shows an accumulation of the angiogenic cells proximal to A β deposits (Fig. 1G). In the human samples, we have counted and identified the position of the $\text{I}\alpha\text{v}\beta\text{3}^+$ cells and A β plaques in five AD cases (Fig. 2A, B), quantify the p-TAU load (Fig. S1C), and correlated those parameters, showing a clear correlation between A β deposits and $\text{I}\alpha\text{v}\beta\text{3}^+$ cells (Fig. 2C and Fig. S1C). In addition, we have now added five control cases (B&B 0-I) and quantified the number of $\text{I}\alpha\text{v}\beta\text{3}^+$ cells (Fig. 2C). Finally, we have calculated the geodesic distances between each $\text{I}\alpha\text{v}\beta\text{3}^+$ cell and the closest A β plaques in the five AD cases. To estimate if the $\text{I}\alpha\text{v}\beta\text{3}^+$ cells were aleatorily distributed in the brain parenchyma or associated with A β plaques, we performed computer-assisted simulations where the positions of the A β plaques in the images were conserved and the coordinates of $\text{I}\alpha\text{v}\beta\text{3}^+$ cells were randomly generated 500 times. In all individual cases studied, the real shortest distance between $\text{I}\alpha\text{v}\beta\text{3}^+$ cells and A β plaques was always smaller than in the random simulations (Fig. 2D and Fig. S1D), and the experimental global average of shortest distances was smaller than the averaged distances in the simulations (Fig. 2E), strongly supporting that $\text{I}\alpha\text{v}\beta\text{3}^+$ cells are not randomly distributed but associated with A β plaques in the human brain.

1.4. The significance of the Evans blue (EB) experiments to assess microvascular perfusion is also unclear. The assumption is that the lack of EB fluorescence reflects non-patent vessels, but global cerebral perfusion is reduced in these mice and there are also microvascular occlusions by circulating leukocytes (Nat. Med. 21, 880; 2015). More quantitative methods to assess microvascular perfusion around the plaques and elsewhere would be needed.

We thank the reviewer for the suggestion. We have now quantified the EB in distal and proximal regions to A β plaques and show that reduced perfusion is concentrated around A β plaques in AD mouse models (Fig. 5A). In addition, we have estimated perfusion using a red cell marker (TER119) and quantified it in wild-type mice and in distal and proximal regions to A β plaques (Fig. 5A). Interestingly, no differences were found between wild-type and AD distal areas (Fig. 5A). It is worth noting that we quantified the areas containing red cells (perfused) with both techniques. However, previous studies concentrated in perfused vessels and quantified the reduction in blood flow. Both results are perfectly complementary and further work will be required to study the relation between both events (reduced blood flow and areas lacking perfusion –and vessels, see other figures of our article–). Indeed, we think that foci where the blood vessels are loss could contribute to increase the brain resistance to the blood flow and accumulation of leukocytes, helping to explain previous findings. However, at this point, that is rather speculative and we have not included any comment regarding this possibility in the manuscript (see also our response to commentary 1.9.).

1.5. In the experiments with AAV-BR1 in WT mice, the efficacy in gamma secretase inhibition and the cerebral endothelial selectivity of the effect needs to be demonstrated. It is unclear what the reduction in the pattern of EB stain and patchy IB4 stain mean.

We thank the reviewer for the suggestion. We have used the original AAV-BR1-Cre produced by the laboratory of Dr. Jakob Körbelin (University of Eppendorf), which have been used in other articles (see^{5–7}). In addition, we have produced our own vectors at the Spanish National Cardiovascular Research Centre (CNIC) and tested that the virus targeted the brain when injected in the tail vein (see Reviewers' Fig. 4). Finally, to show the efficiency of recombination and the targeting of brain endothelial cells, we have developed a qPCR assay to test brain excision of the *Psen1*^{Floxed} allele (Fig. 4B), quantified the total level of *Psen1* mRNA by ISH (Fig. S4A), and the levels of endothelial *Psen1* mRNA (Fig. 4C, D). Regarding EB staining, in our hands, EB is quite good in quantifying perfusion between different areas of the same brain (for instance, distal and proximal to A β plaques), but less accurate to compare between brains. To complement our study, we have quantified perfusion using EB and TER119 in the VaS-like areas and in normal vessels, showing a clear reduction of perfusion in the areas that contain IB4⁺ vascular abnormalities (Fig. S4B).

1.6. It is also unclear how these results targeting PS1 deletion to endothelial cells relate to those in APP-PS1 mice in which the PS1 mutation is directed preferentially to brain cells with the prion protein promoter.

We apologize for not being able to convey that information. We agree with the reviewer that the loss of PS1 activity cannot be ascribed to the PS1 mutation in the *APP-PSEN1* AD mouse model, as it is expressed from the mainly neuronal prion promoter⁹. Indeed, that is why we used the *APP*₇₅₁*SL* model that does not express any mutated form of PS1. In this model, the VaS were identical to the ones found in *APP-PSEN1* (Fig. 3G, H), ruling out any contribution of the mutated form of PS1 in neurons to VaS formation. We have now included a full paragraph to better explain this point (Page 10, lines 14 to 25). In addition, we had in the discussion a paragraph suggesting a possible mechanism for the reduced endothelial lateral inhibition found around A β plaques (Page 22, lines 15 to 18).

1.7. Concerning the role of microglia, Iba1 not a specific microglial marker and a role of brain macrophages cannot be ruled out.

We fully agree with the reviewer. Unfortunately, the main marker to differentiate microglial cells from brain macrophages is TMEM119⁹, which, however, is strongly downregulated around A β plaques¹⁰. We have tested this marker (data not shown) and the labelling of innate immune cells around A β plaques, as expected, is strongly reduced. To include the possibility of a role of brain macrophages, we have added a paragraph in the discussion (**Page 22, lines 1 to 5**). However, we find it unlikely due to the modest role of those cells in AD mouse models¹⁰ and also based in our transcriptional analysis of activated microglia from the *APP^{751SL}* mouse model (CD11b/CD45/CLEC7a triple immunoreactive cells), where we did not observe any enrichment in peripheral macrophages markers².

1.8. The reported phagocytic activity may be related to tissue remodeling caused by inflammation around the plaques, rather than specifically targeting the vessels.

Again, we thank the reviewer for the suggestion. It is something that we have considered for a long time, however, the fact that we can induce the same vascular abnormalities by targeting *Psen1* in endothelial cells indicates a primary role of those cells in inducing vessels regression (**Fig. 4**). To test the contribution of inflammation, we have now included in the article a model of experimental autoimmune encephalitis (EAE), where immunization with a myelin peptide induces a strong innate immune reaction around blood vessels (**Fig. S3G**). However, we did not observe any structures that could resemble the VaS in 103 autoimmune foci from 15 mice (**Page 10, lines 2 to 10**).

1.9. The results of the experiment using ICP as a surrogate for ISF bulk flow/CSF clearance are difficult to interpret considering the wide variety of structural and functional factors that can alter ICP.

We agree with the reviewer and we have now removed this part from the new version of the manuscript. Correspondingly, we have also reduced the comments about clearance of A β by the vascular system, as it is not the main topic of our work. Those experiments will be complemented and published elsewhere.

1.10. Similarly, the data with sorafenib are difficult to interpret due to wide variety of pharmacological effects of the drug (c-raf, NF κ b, etc.). For example, it has been previously shown that the drug is protective in APP mice, an effect attributed to suppression of inflammation (e.g, Neuroscience. 2009;162:1220). Finally, the cognitive impact of the treatment needs to be assessed and the ratio between A β 40 and 42 determined.

We agree with the reviewer, so we have moved the sorafenib data to the supplementary information (**Fig. S6**), complemented it with the determination of A β ₁₋₄₀ and A β ₁₋₄₂ levels by ELISA (**Fig. S6C**), the quantification of vessel normalization using PDGFR β (**Fig. S6D**), and the estimation of the cognitive impact of the treatment (**Fig. S6E**). We have also clarified in the text that this treatment is not optimal to prevent angiogenesis failure in AD, as it only prevents the initiation of angiogenesis. An ideal treatment should allow the angiogenesis to perfuse the A β plaques (**Page 16, lines 3 to 21**).

Reviewer #2 (Remarks to the Author):

The authors report that A-beta deposits are surrounded by abnormal vascular structures. These structures contain VEGF but angiogenesis is impaired are there are some signs of vasoregression.

The study addresses a very interesting finding in Alzheimer brains which others have reported before: vascular alterations around A-beta deposits. This study now aims to explore the underlying mechanisms. The authors present many interesting and provoking findings. The overall concept is pretty intriguing. Nevertheless the data do not (yet) justify the multiple conclusions drawn by the authors. The study is still preliminary in many aspects.

We thank the reviewer for the time employed in the evaluation and for his/her commentaries that have, without doubt, improved the quality and readability of our manuscript. We have now performed extensive experimental work to enhance our previous version and to focus the manuscript around the angiogenic problems associated with A β plaques.

Major points:

2.1. I agree that the structures and the gene signatures are somehow reminiscent of Dll4/Notch1 or Rbpj deficient mouse models. However, to link the pathology to endothelial Notch signaling alterations a much deeper analysis would be required. Why do the authors use a PS1/2 gene deletion model instead of a Dll4/Notch model. Presenilins do not only cleave Notch receptors but have several additional substrates.

We agree that NOTCH models will be more accurate to understand the NOTCH signaling in AD. However, the PS loss of function mutations are the one linked with fAD and, from an AD perspective, is more relevant to characterize the failure of this pathway. To link our findings to NOTCH, we have been able to generate a new mouse model where a NICD transcriptional reporter drive the expression of a nuclear Venus¹¹. By breeding those mice with an AD mouse model, we have shown that the endothelial NICD activity is strongly reduced around A β plaques (**Fig. 3J, K**). In addition, we have also shown the accumulation of *Plaur* mRNA around A β plaques. To our knowledge, one of the few markers with IB4 of tip cells described so far¹² (**Fig. 3D**). We apologize for not showing in the reviewed version of the manuscript a genetic model with

manipulation of the NOTCH activity. It will take around 1.5–2 years to have the mice of the appropriated genotype and age in adequate numbers to quantify the role of NOTCH (in a non-pandemic time scale). The experiments are running and they will be published elsewhere.

2.2. Also the description of the AAV mediated PS1/2 deletion in the brain vasculature appears superficial. Most importantly, it remains unclear how specific this procedure is for targeting the genes only in endothelial cells. Moreover, does this cause increased permeability, BBB breakdown, edema, etc. It is amazing to see that the quiescent brain vasculature can be altered in such a dramatic manner. It would be important to analyze how fast the vascular changes occur. Also, what is the percentage of endothelial cells targeted by the AAV and how many of these cells are involved in the formation of vascular lesions. Do the endothelial cells in such lesion proliferate? Are there signs of endothelial-to-mesenchymal transformation? Is there a dropout of pericytes only at lesions?

We thank the reviewer for the suggestion. We have used the original AAV-BR1-Cre produced by the laboratory of Dr. Jakob Körbelin (University of Eppendorf), which have been used in other articles (see⁵⁻⁷). In addition, we have produced our own vectors at the Spanish National Cardiovascular Research Centre (CNIC) and tested that the virus targeted the brain when injected in the tail vein (see **Reviewers' Fig. 4**). Finally, to show the efficiency of recombination and the targeting of brain endothelial cells, we have developed a qPCR assay to test brain excision of the *Psen1^{Floxed}* allele (**Fig. 4B**), quantified the total level of *Psen1* mRNA by ISH (**Fig. S4A**), and the levels of endothelial *Psen1* mRNA (**Fig. 4C, D**).

We have also quantified the perfusion of altered vessels using EB and TER119 (a red blood marker; **Fig. S4B**), two different markers of the loss of the BBB at the lesions: PDGFR β and AQP4 (**Fig. 4F, G**), and the absence of proliferation around VaS like structures (Ki67; **Pages 13, lines 23 to 25 and 14, lines 1 and 2**), as expected if endothelial cells were not differentiating towards proliferating stalk cells.

We agree with the reviewer that is a strong phenotype and quite unexpected. However, the adult vasculature is not quiescent, as it has been demonstrated that angiogenesis can be stimulated in brain areas following intense behavioral training or stimulation^{13,14}, suggesting that a basal angiogenesis is taking place associated with brain function. In agreement, we also observed few $\alpha v\beta 3^+$ angiogenic cells in the brain of wild-type mice (**Fig. 1G**), suggesting that adult angiogenesis will deserve further work. Of note, we only observed VaS like structures in striatum and the hippocampus and really few in the cortex of AAV-BR1-Cre-injected mice, although we observed deletion of the *Psen1* allele in all those structures (**Fig. 4**). These data suggest that angiogenesis may be more frequent in those structures than in the cortex, facilitating the double hit required for VaS formation: (i) angiogenesis initiation and (ii) loss of PS function. Interestingly, quantification of two different time points after the injection of the virus (19 and 60 days) suggests that the VaS like structures are a consequence of vessel regression (**Fig. 6A**), as we have less cells with excised *Psen1* allele at 60 days (**Pages 12, lines 22 to 25 and 14, line 1**), suggesting its removal by microglial phagocytosis (**Fig. 6A**). This temporal study also suggests that the phenotype requires time to reach full expressivity (60 days) (**Page 13, lines 12 to 15**).

2.3. The authors have crossed mice with the Cdh6-Cre-ERT2. Why don't they use this cre to delete PS1/2 in blood vessels in adult mice instead of the AAV approach?

Chd5-Cre::ERT2 drives Cre expression in all the vascular system, something that could alter the homeostasis of several organs. In such scenario, it will be difficult to ascribe a role of the PS function to brain vasculature. For instance, alteration of the endothelial NOTCH system in the adult heart is associated with metabolic and functional changes¹⁵ that could influence in the natural history of A β accumulation in the mouse models.

2.4. The use of Sorafenib gives interesting results. However, this RTK inhibitor has numerous substrates and is not very specific for inhibiting angiogenesis. The authors should use a VEGF inhibiting approach what would also fit better to the findings about increased VEGF levels. It also needs to be noted that VEGF is not only an angiogenesis stimulus but also a permeability factor. Altering permeability alone could also explain some of the findings.

We agree with the reviewer, so we have moved the sorafenib data to the supplementary information (**Fig. S6**), complemented it with the determination of A β_{1-40} and A β_{1-42} levels by ELISA (**Fig. S6C**), the quantification of vessel normalization using PDGFR β (**Fig. S6D**), and the estimation of the cognitive impact of the treatment (**Fig. S6E**). We have also clarified in the text that this treatment is not optimal to prevent angiogenesis failure in AD, as it only prevents the initiation of angiogenesis, when an ideal treatment should allow the angiogenesis to perfuse the A β plaques (**Page 16, lines 3 to 21**). A similar partial result should be produced with the inhibition of VEGF signaling, as we will not improve angiogenesis and only block the deleterious effect of the halted process. We apologize for not performing the chronic VEGF inhibition, but we did not have enough mice to complete the characterization of the sorafenib model (as asked for by another reviewer; point **1.10.**) and doing a full study with a VEGF inhibitor. In addition, the cost of chronic treatment of a rather big number of mice will be very challenging. We agree with the reviewer that VEGF acting as a permeability factor could explain some of the results that we are presenting. In our view, permeabilization is another step in the angiogenic process, required for tip cells extrusion and local growing of new vascular branches. The fact that there is accumulation of tip cells markers nearby the A β plaques is

indicative that angiogenesis has started and is proceeding far beyond the permeabilizing role of VEGF. In any case, under such angiogenic halted conditions, VEGF could potentiate the accumulation of IB4 deposits through vascular permeabilization. We have now added a sentence to the discussion to highlight this possibility (**Page 21, lines 2 and 3**).

2.5. If it is really dysregulated Notch signaling in endothelial cells that explains the vascular alterations, why do the authors not attempt to rescue it with a Notch gain of function approach.

To link our findings to NOTCH, we have been able to generate a new mouse model where a NICD transcriptional reporter drive the expression of a nuclear Venus¹¹. By breeding those mice with an AD mouse model, we have shown that the endothelial NICD activity is strongly reduced around A β plaques (**Fig. 3J, K**). In addition, we have also shown the accumulation of *Plaur* mRNA around A β plaques. To our knowledge, one of the few markers with IB4 of tip cells described so far¹² (**Fig. 3D**).

As explained before, we have focused on the role of PS due to their relevance in AD (see also response to comment 2.1.). We apologize for not showing in the reviewed version of the manuscript a genetic model with manipulation of the NOTCH activity. It will take around 1.5 – 2 years to have the mice of the appropriated genotype and age in adequate numbers to quantify the role of NOTCH (in a non-pandemic time scale). The experiments are running and they will be published elsewhere.

2.6. There is absolutely no explanation in this paper why A-beta could interfere with Delta Notch signaling in the endothelium.

Again, we apologize for our lack of clarity. In the previous version we had a paragraph in the discussion section where we discussed the possibility of A β acting as inhibitor of the activity of the gamma-secretase. That is not our idea, but whose proposed in 2007 by Prof. Jie Shen¹⁶. The underlying hypothesis is that A β , as the biochemical product of the gamma-secretase reaction, can inhibit the activity of the enzyme when found at high concentration. To gain clarity in the reading, we have introduced that concept in the result section when presenting the AAV-BR1-Cre model (**Page 12, lines 14 to 16**).

Reviewer #3 (Remarks to the Author):

Rosales-Nieves & Alvarez-Vergara et al. perform experiments suggesting that there are more angiogenic markers near amyloid-beta plaques in APP transgenic mice that are potentially derived from astrocytes. Floxing out PS1 from endothelial cells appears to increase an endothelial marker IB4 that may be non-productive. The findings are interesting and contribute to a new idea related to how A β may impact the brain by potentially inducing non-productive/functional vascular components. However, without functional data, it is a bit difficult to determine whether this marker does indeed represent functionless vessels or pathological angiogenesis. There also appears to be Iba1+ microglia that the authors are calling “vascular scars” near the amyloid-beta plaques that are engulfing vascular material. Interestingly, sorafenib (a kinase inhibitor) that decreases angiogenesis decreases amyloid-beta plaque load. They suggest that this may be due to enhanced A β clearance from the parenchyma though there is no data to directly support this mechanism. The authors also make other conclusions in this manuscript, but many experiments with appropriate controls are suggested to reach those conclusions.

Overall, the findings here have an opportunity to make new contributions to the literature but a number of issues need to be addressed.

We thank the reviewer for the time employed in the evaluation and for his/her commentaries that have, without doubt, improved the quality and readability of our manuscript.

Major points:

3.1. 1) Most of the statistics in this paper are not performed in the appropriate fashion. Each plaque or cell is counted as an individual biological replicate, when instead each mouse or human should represent a single sample. For instance, Fig 1E suggests that there are n=2995 human brain samples, but there are in fact only n=4 samples. These analyses inflate the sample size and bias statistics towards being significant. The authors need to re-calculate their statistics based on individual samples, and they should NOT be representing each cell/image/brain section as an individual biological replica. Furthermore, all images should be quantified and analyzed with appropriate statistics.

We have now reanalyzed all the quantifications and the results are presented as averaged data per mouse/human sample. In the figure 1E (**Fig. 2** in this new version), we think that is important to show, in addition to the averaged results per human sample (**Fig. 2E**), the individual results per case (**Fig. 2D** and **Fig. S2D**). We have also quantified all the images as requested (see below).

3.2. 2) Fig 1:a. Fig 1D-G: The methods for the quantification need to be described. For example, in Fig 1C, were non-microglial cells assumed to be astrocytes? How many regions were imaged per mouse? Which regions? Etc. How was cell density calculated in Fig 1E?

We have now improved the description of the quantification performed in the online methods section (**Online Methods section –OM– Page 17, lines 6 to 25**). Specifically, in **Fig.1C**, we have counted double labelled astrocytes with *Vegfa* ISH and GFAP (**Fig. 1C**) and microglia with *Vegfa* ISH and IBA1 (**Fig. 1D**) from two

independent experiments, no cell identity was assumed. In all experiments, cortical regions (information now included in figure legends) were analyzed at least otherwise specified in the figure legend. Between 5 and 10 regions were analyzed per mice in each quantification and we have now included this information in the online methods section (**OM Page 17, lines 9 to 10**). Density was calculated by dividing the number of $\alpha\text{v}\beta 3^+$ cells by the area examined.

3.3. b. Fig 1F: Given that there are human brains available, the authors should additionally reproduce in their hands whether there are more Integrin $\alpha\text{v}\beta 3^+$ cells in humans with AD vs. controls, in addition to the mouse data where they show that there are more Integrin $\alpha\text{v}\beta 3^+$ cells in APP/PS1 mice vs. controls. We thank the reviewer for the suggestion, we have included one more AD brain and 5 control cases (Braak & Braak 0-I) to the analysis. The quantification indicates a clear increase in the number of $\alpha\text{v}\beta 3^+$ cells between AD and Control cases (**Fig. 2C**).

3.4. c. If the authors want to conclude that angiogenesis is concentrated around plaques, a necessary experiment is to compare their data to how much angiogenesis exists outside of plaques. They do this in Fig 4A where there is a distal vs proximal plaque comparison; they should perform a similar analysis for this figure. Without this comparison, there might appear to be more angiogenesis around plaques when there is actually just an upregulation of angiogenesis everywhere in the brain. Furthermore, with the current set of experiments, the authors cannot conclude that pathological angiogenesis is “initiated” around plaques. This would require experiments with mice taken multiple timepoints starting before plaque onset.

We have now included the quantification of the number of $\alpha\text{v}\beta 3^+$ cells in wild-type and in distal and proximal regions to A β (**Fig. 1G**). The results clearly show an accumulation of those cells nearby A β plaques. We have also included a temporal analysis of the VaS phenotype, showing no vascular defects in mouse without A β plaques (3-month-old) and only associated with A β plaques in mice with incipient pathology (5-month-old) (**Fig. S3H**).

3.5. 3) Fig 2: a. The age of APPPS1 mice vary a lot here (8, 10, 12, 18 mo). Please justify the different ages and why a single age was not used. Some phenotypes seen may be age-dependent.

We have added a supplementary table with all the ages used (**Table S1**). We have mainly used 8-month-old mice, an age with abundant A β pathology in the two mouse models used (*APP-PSEN1* and *APP_{751SL}*). For hypoxia determination (**Fig. 1B**) and microarray analysis (**Fig. 3B**), we used older mice to have a full penetrant phenotype. In particular, in **Fig. 3B** (old Fig. 2), we isolated total endothelial cells, so we choose an age where the pathology was highly extensive, to avoid the dilution with non-altered distal endothelial cells. In any case, we now show a *Plaur* mRNA ISH in 8-month-old mice that validates the molecular changes observed at later stages (**Fig. 3D**). It is worth noting that A β plaques can be older in a younger mouse and *vice versa*, as the age of every individual plaque cannot be determined. In any case, the vast majority of experiments are performed with 8-month-old mice as stated.

3.6. b. Staining for amyloid-beta plaques in some way (rather than relying on their autofluorescence) is crucial for their identification.

We have relayed in Thio-S staining in the vast majority of experiments to localize the A β plaques. However, in some cases we need that channel to follow a specific marker. In those cases, a trained eye has not difficulty to localize A β plaques based in their autofluorescence in the DAPI channels. We agree that the images presented not always identified A β plaques, however, every plaque quantified was rigorously screened in the confocal microscopy to show the presence of the deposit. The images presented are sometimes Z confocal planes that cover only 0.5 μm , and therefore are not representative of the deposits. A sentence has been added to the online methods section to clarify that point (**OM Page 17, lines 10 to 11**).

3.7. c. Fig 2D: The authors claim that there is reduced reperfusion in 12 mo APP mice compared to 8 mo APP mice, however a quantification is needed for the Evans Blue staining, and this quantification needs to be compared to a WT control to make this claim.

We apologize for the overstatement. We have now removed this sentence, as is not the focus of our paper. Following the indication of the reviewer, we have quantified the EB in distal and proximal regions to A β plaques and shown that reduced perfusion is concentrated around A β plaques in AD mouse models (**Fig. 5A**). In addition, we have estimated perfusion using a red cell marker (TER119) and quantified it in wild-type mice in distal and proximal regions to A β plaques (**Fig. 5A**). Interestingly, no differences were found between wild-type and AD distal areas (**Fig. 5A**).

3.8. d. Fig 3G also needs comparisons against a WT treated mouse.

We thank the reviewer for the suggestion. We understand the reviewer is asking for **Fig. 2G** in the previous version, **Fig. 3H** in the new version. A WT quantification has been added to **Fig. 3H**.

3.9. e. It would be worthwhile to show that the lack of Evans Blue staining correlates with stronger IB4 staining, which would suggest that indeed these vessels are less productive.

We thank the reviewer for the suggestion. As we observe a loss of both perfusion (**Fig. 5**) and the vessel themselves (**Fig. 5** and **Fig. S5**), we have now quantified the amount of laminin marker in WT, distal,

proximal, and IB4⁺ regions. Interestingly, the loss of the vessel was associated with the IB4 staining (**Fig. 5**).

3.10. f. It is surprising that in the microarrays, there was no change in IB4 or VEGF genes. To corroborate the claim that angiogenesis is non-productive around plaques, staining or in situ hybridization of microarray genes (Fig 2B) around plaques would support this claim.

We apologize, for not making this point clear. IB4 is a lectin from a plant (a protein able to recognize specific glycosylated residues in specific proteins) (**Page 9, lines 11 to 13**), so there is not gene encoding for it in the mouse genome. Unfortunately, we do not know the endothelial protein that is marked by it. Regarding *Vegfa*, endothelial cells do not suffer hypoxia, as they are conducting the blood. It is the nearby cells that, under low oxygen, express *Vegfa* to induce the growth of new vessels. As the microarray experiment was done with endothelial cells, it is perfectly normal that *Vegfa* gene was not found upregulated. We have now provided another manuscript as complementary information to this review (March-Diaz *et al.*, under second review in Nat. Aging), where you can observe that in activated microglia (the one found nearby A β plaques), *Vegfa* mRNA levels are increased compared to WT microglia (**Reviewers' Fig. 1**).

We have also included the validation of the microarray using ISH as suggested. We have now stained and quantify the *Plaur* mRNA levels in WT, distal, and proximal regions to A β plaques (**Fig. 3D**).

3.11. 4) Fig 3: a. Conditional knockout of PS1 in endothelial cells using AAV-BR1-cre shows an increase in IB4 coverage, and this needs to be quantified compared to AAV-controls.

We thank the reviewer for the suggestion. We have now quantified the IB4 coverage (**Fig. 4E**).

3.12. 5) Fig 4 and Fig S3: a. The data in Fig 4A is convincing in showing that there is laminin loss proximal to plaques. A great control would be to show the expected level of laminin in a WT mouse with no pathology in just the distal regions.

We thank the reviewer for the suggestion. We have now quantified the amount of laminin marker in WT, distal, proximal, and IB4⁺ regions. Interestingly, the loss of the vessel was associated with the IB4 staining (**Fig. 5**).

3.13. b. To implicate that there are less PDGFRbeta⁺ pericytes, CD31⁺ endothelial, and astrocytic end-feet, quantification is needed (Fig S3A, B, C; Fig 4B).

We thank the reviewer for the suggestion. We have now quantified the amount of those markers in WT, distal, and proximal regions (**Fig. 5** and **Fig. S5**).

3.14. 6) Fig 5 and Fig. S4: a. Both figures need quantification. How many Iba1⁺ microglia are there around blood vessels distal or proximal from plaques? Which regions were imaged?

We thank the reviewer for the suggestion. We have now quantified the contacts between microglia and tdTomato⁺ vessels in WT, distal, and proximal regions (**Fig. 6C**).

3.15. b. If the authors claim that the tdTomato⁺ microglial pouches are phagocytic microglia, then a CD68 stain (or other marker for phagocytic process) for phagocytic microglia should be performed. The authors should be able to see vascular components inside the lysosomes of microglia.

We thank the reviewer for the suggestion. We have now stained microglia using CD68. We observed that almost all the tdTomato⁺ microglial pouches are also CD68⁺ (**Fig. 6D**).

3.16. c. To show that microglia (and not other cell types like astrocytes) are surrounding blood vessels to create vascular scars, a comparison between the number of microglia or astrocytes surrounding blood vessels is necessary.

We thank the reviewer for the suggestion. To quantify the astrocytes is not an easy task. GFAP is labelling the projections but does not clearly identify the cells that emit them, making the quantification quite difficult. To our knowledge, the best marker to study the interaction between astrocytes and blood vessels is AQP4, which specifically labels the astrocytic end-feet that contact with endothelial cells. We have quantified the AQP4 staining in WT, distal and proximal regions and found a clear decrease around A β plaques (**Fig. 6C**). We could not observe any tdTomato⁺ material inside astrocytic projections (either GFAP or AQP4⁺). However, we have included a paragraph in the discussion section letting open the possible contribution of those cells to the vascular regression (**Pages 21, lines 24 and 25 and 22, line 1**).

3.17. d. The thickening of microglial projections also needs to be quantified.

Microglia staining is a continuum around the vessels (**Fig. 6C**), making almost impossible to differentiate projections from one cell to another. We have preferred to remove the sentence better than provide a probably erroneous quantification.

3.18. e. It would be worthwhile to discuss the relationship between astrocyte-derived VEGF and its relationship with microglia surrounding vessels and colocalizing with IB4.

We now added a paragraph in the discussion section highlighting the role of those cells in the induction of microglia (**Pages 21, lines 24 to 25 and 22, line 1**).

3.19. 7) Fig 6: a. It is very interesting that there is less intracranial pressure in APP transgenic mice compared to WT mice, potentially indicating less ISF bulk flow. To prove that sorafenib does indeed improve clearance and bulk flow, the authors should show that ISF bulk flow is restored with ICP.

We apologize, reviewer #1 was very critical with the interpretation of those data (see our response to point 1.9.) and we have removed the data from the manuscript. Correspondingly, we have also reduced the comments about clearance of A β by the vascular system, as it is not the main topic of our work. Those experiments will be complemented and published elsewhere.

3.20. b. IB4 was used as a proxy for angiogenic endothelial cells. To confirm that more endothelial cell expression is indicative of angiogenesis, a classic marker for angiogenesis should also be used, such as Tie2.

We thank the suggestion of the reviewer. We have searched for other markers of tip cells, but there are really few of them in the literature that unequivocally detect them. We agree with the reviewer that Tie2 signaling is activated in tip cells, however, Tie2 expression is not confined to angiogenic cells, but found all along the endothelium (see **Reviewers' Fig. 5**; see for instance¹⁷). As we understand the relevance of this experiment, we used *Plaur*, one of the genes found as upregulated in the *Dll4*^{+/-}_UP signature in the retina and in AD endothelial cells, as a marker of tip cells¹² (**Fig. 3D**).

Minor points:

3.21. 1) State the sex of the mice for all experiments.

We are sorry for the inconveniences. We have now included a supplementary table containing that information (**Table S1**).

3.22. 2) Fig 2: Clarify why *Dll4*^{+/-} mice were used (i.e. angiogenesis is down regulated in these mice).

Indeed, *Dll4*^{+/-} mice have a problem to proceed with angiogenesis, not a down regulation of angiogenesis. Those mice accumulate tip cells in the developing retina due to a failure in the lateral inhibition, causing non-productive angiogenesis¹⁸. The phenotype observed is quite similar to the one observed nearby A β plaques and also associates with loss of perfusion¹⁸⁻²¹. The rationale for using *Dll4*^{+/-} was explained in the introduction (**Page 5, lines 2 to 8**). To facilitate reading, we have now included a sentence in the result section (**Page 9, line 3**).

3.23. Introduce APPSL+ mice and justify why these mice were used in addition to the APPPS1 mice.

We apologize. The description and justification can be now found in the text (**Page 10, lines 14 to 25**).

3.24. 3) State frequency/duration of sorafenib treatment.

Again, we apologize for the mistake. The data can now be found in the text (**Page 16, lines 6 and 7**), the online methods (**OM, Page 11, line 5**), and in the supplementary figure legend.

References

1. Christie, R. H. *et al.* Growth arrest of individual senile plaques in a model of Alzheimer's disease observed by in vivo multiphoton microscopy. *J. Neurosci.* **21**, 858–864 (2001).
2. March-Diaz, R. *et al.* Systemic and Local Hypoxia Synergize Through HIF1 to Compromise the Mitochondrial Metabolism of Alzheimer's Disease Microglia. *SSRN Electron. J.* (2019). doi:10.2139/ssrn.3443013
3. Vanlandewijck, M. *et al.* A molecular atlas of cell types and zonation in the brain vasculature. *Nature* **554**, (2018).
4. Zhang, Y. *et al.* An RNA-sequencing transcriptome and splicing database of glia, neurons, and vascular cells of the cerebral cortex. *J. Neurosci.* **34**, 11929–11947 (2014).
5. Dogbevia, G. K. *et al.* Gene therapy decreases seizures in a model of Incontinentia pigmenti. *Ann. Neurol.* **82**, 93–104 (2017).
6. Tan, C. *et al.* Endothelium-Derived Semaphorin 3G Regulates Hippocampal Synaptic Structure and Plasticity via Neuropilin-2/PlexinA4. *Neuron* **101**, 920-937.e13 (2019).
7. Yousef, H. *et al.* Aged blood impairs hippocampal neural precursor activity and activates microglia via brain endothelial cell VCAM1. *Nat. Med.* **25**, 988–1000 (2019).
8. Jankowsky, J. L. *et al.* Mutant presenilins specifically elevate the levels of the 42 residue beta-amyloid peptide in vivo: evidence for augmentation of a 42-specific gamma secretase. *Hum. Mol. Genet.* **13**, 159–70 (2004).
9. Bennett, M. L. *et al.* New tools for studying microglia in the mouse and human CNS. *Pnas* **113**, 1525528113- (2016).
10. Keren-Shaul, H. *et al.* A Unique Microglia Type Associated with Restricting Development of Alzheimer's Disease. *Cell* **169**, 1–15 (2017).
11. Nowotschin, S., Xenopoulos, P., Schrode, N. & Hadjantonakis, A. K. A bright single-cell resolution live imaging reporter of Notch signaling in the mouse. *BMC Dev. Biol.* **13**, (2013).
12. del Toro, R. *et al.* Identification and functional analysis of endothelial tip cell enriched genes. *Blood* **116**, 4025–4033 (2010).
13. Bengoetxea, H., Argandoña, E. G. & Lafuente, J. V. Effects of visual experience on vascular endothelial growth factor expression during the postnatal development of the rat visual cortex. *Cereb. Cortex* **18**, 1630–1639 (2008).
14. G. Argandona, E. *et al.* Vascular Endothelial Growth Factor: Adaptive Changes in the Neuroglialvascular Unit. *Curr. Neurovasc. Res.* **9**, 72–81 (2012).
15. Jabs, M. *et al.* Inhibition of Endothelial Notch Signaling Impairs Fatty Acid Transport and Leads to Metabolic and Vascular Remodeling of the Adult Heart. *Circulation* **137**, 2592–2608 (2018).
16. Shen, J. & Kelleher, R. J. The presenilin hypothesis of Alzheimer's disease: evidence for a loss-of-function pathogenic mechanism. *Proc. Natl. Acad. Sci. U. S. A.* **104**, 403–9 (2007).
17. Wong, A. L. *et al.* Tie2 Expression and Phosphorylation in Angiogenic and Quiescent Adult Tissues. *Circ. Res.* **81**, 567–574 (1997).
18. Noguera-Troise, I. *et al.* Blockade of Dll4 inhibits tumour growth by promoting non-productive angiogenesis. *Nature* **444**, 1032–1037 (2006).
19. Suchting, S. *et al.* The Notch ligand Delta-like 4 negatively regulates endothelial tip cell formation and vessel branching. *Proc. Natl. Acad. Sci.* **104**, 3225–3230 (2007).
20. Lobov, I. B. *et al.* Delta-like ligand 4 (Dll4) is induced by VEGF as a negative regulator of angiogenic sprouting. *Proc. Natl. Acad. Sci.* **104**, 3219–3224 (2007).
21. Ridgway, J. *et al.* Inhibition of Dll4 signalling inhibits tumour growth by deregulating angiogenesis. *Nature* **444**, 1083–1087 (2006).
22. Orre, M. *et al.* Isolation of glia from Alzheimer's mice reveals inflammation and dysfunction. *Neurobiol. Aging* **35**, 2746–60 (2014).

Reviewers' Figures

Reviewers' Figure 1 from
March-Diaz *et al.*,

RFig. 1 | HIF1-mediated transcription is activated in AβAM. **a, b**, *In situ* hybridization (ISH) of *Hif1a* mRNA (brown) combined with immunohistochemistry for microglia (IBA1, green), and nuclear (DAPI; blue) staining in coronal brain sections of 8-month-old *APP-PSEN1*^{+/+} mice proximal (**a**) and distal (**a, b**) to Aβ plaques (yellow asterisks in **a**). Red arrowheads indicate microglial cells proximal to Aβ plaques with strong *Hif1a* expression and yellow arrowheads depict microglial cells not associated with Aβ deposits. Scale bars are 20 μm. Right graph, quantification of the microglial (IBA1⁺) ISH *Hif1a* mRNA signal in wild-type mice (WT; grey dots) and in distal (light blue dots) and proximal (blue dots) areas to Aβ plaques from 8-month-old *APP-PSEN1*^{+/+} mice ($n = 4$ mice; ANOVA, post hoc Tukey's test ** $p < 0.01$). **c**, Transcription Factor Enrichment Analysis (TFEA) of *APP-PSEN1*^{+/+} microglial transcription²². Each dot in the volcano plot represent an individual CHIP-seq experiment. **d**, Volcano plot (right panel) showing the genes included in the hypoxia/HIF1-induced microglial module (HMM) (red dots; $p < 0.01$, LogFC > 0.5, red rectangle). **e**, Primary microglial cell cultures from *Cx3cr1-Cre::ERT2*^{+/+}; *Hif1a*^{Flox/Flox} mice were either treated with vehicle (C; grey bar) or tamoxifen (T; 100 nM; magenta bar) and the effect

on HIF1 α expression was assayed by qRT-PCR (left panel; $n = 7$; * $p < 0.05$ Student's t -test) and by western blot (right panel) in normoxia (N: 21% O₂, 24 h) and after DMOG (D: 0.1 mM, 24 h) treatment. **f**, Primary microglial cell cultures from *Cx3cr1-Cre::ERT2/+; Hif1a^{Flox/Flox}* mice were either treated with vehicle, DMOG (D: 0.1 mM, 24 h), tamoxifen (T: 100 nM), or tamoxifen and DMOG and the mRNA fold change DMOG *versus* vehicle (D; blue bars) and DMOG-tamoxifen *versus* tamoxifen (D+T; magenta bars) are represented. *Hmbs* levels were used as housekeeping control and dotted lines represent no induction ($n = 3 - 5$; * $p < 0.05$; ** $p < 0.01$ paired-Student's t -test). **g**, Gene set enrichment analysis (GSEA) of *APP₇₅₁SL/+* DAM *versus* (vs) wild-type (WT) 12-month-old (mo) microglia. Heat maps of up to the top 30 ranking leading edge genes of the Hypoxia/HIF1-induced microglial modules (HMM; left) and MGnD (right) gene sets (GSs). Red symbolizes overexpression and blue down regulation (color shades values can be inferred from **Supplementary tables**). Right table: 15 top GSs with a FWER- p -value less than 0.05 are listed.

Reviewers' Figure 2 from March-Diaz *et al.*,

RFig. 2 | HIF1-mediated transcription in DAM. **a**, (A) Immunocytochemistry of primary cultures of microglia (left panel) and astrocytes (right panel) using antibodies against IBA1 (red) and GFAP (green). Scale bar is 20 μ m. **b**, Relative levels of *Cd33* and *Gfap* mRNAs estimated by qRT-PCR (a.u. arbitrary units) in microglial (M, red) and astrocyte cultures (A, green). Hmbs levels were used as housekeeping control. ** $p < 0.01$ (Student's t-test). Data are represented as mean +

S.E.M. $n = 3 - 4$. **c**, Principal component analysis showing the separation between biological replicates of mouse primary microglial cell cultures exposed to normoxia (N: 21% O₂, 6 h; grey dots) or hypoxia (H: 1% O₂, 6 h; blue dots). **d**, Primary microglial cell cultures were exposed to normoxia (N: 21% O₂, grey bars) or (H: 1% O₂, blue bars) for 24 h and the relative levels of several mRNAs included in the HIF1/hypoxia-induced microglial module (HMM) were estimated by qRT-PCR. *Hmbs* levels were used as housekeeping control ($n = 3 - 6$ cultures; * $p < 0.05$; ** $p < 0.01$; *** $p < 0.001$ Student's *t*-test). **e-h**, Gene set enrichment analysis (GSEA). **e**, GSEA of *APP*₇₅₁*SL*/⁺ (left and middle right) or *MAPT*_{p301S}/⁺ end-stage (ES) (middle left and right) DAM versus (*vs*) wild-type (WT) 12-month-old (mo) microglia. Enrichment plots of the HMM (left) and MGnD (right) GSs. The table contains the 15 top GSs with a FWER-*p*-value less than 0.05. **f**, GSEA of *APP*₇₅₁*SL*/⁺ versus (*vs*) *MAPT*_{p301S}/⁺ end-stage (ES). **g, h**, GSEA *5xfAD*/⁺ DAM vs wild-type (WT) microglia (**g**, upper row MGnD GS; **h**, HMM GS); *SOD1*_{p.G93A}/⁺ DAM vs WT microglia (**g**, left in the lower row); 24mo vs 5mo WT microglia (**g**, right in the lower row).

Vegfa - Mus musculus

Figure B: [Brain data] Average expression in each cluster

Alvarez-Vergara et al. Reviewers' Fig. 3

RFig. 3 | Levels of *Vegfa* mRNA in single cell RNAseq from mouse brains. Top image from⁴. Bottom image from³. FB: fibroblasts; AC: Astrocytes.

Alvarez-Vergara et al. Reviewers' Fig. 4

RFig. 4 | Luciferase signal of AAV-BR1-Luciferase and control mice after injection in the tail vein.

Tek - Mus musculus

Figure B: [Brain data] Average expression in each cluster

Alvarez-Vergara et al. Reviewers' Fig. 5

RFig. 5 | Levels of *Tek* (*Tie2*) mRNA in single cell RNAseq from mouse brains. Top image from⁴. Bottom image from³. EC: endothelial cells.

REVIEWER COMMENTS

Reviewer #1 (Remarks to the Author):

The authors have done an excellent job at addressing the concerns raised in the review. The following questions remain:

1. The cell biological phenomena described here occur quite late in the progression of AD (as shown both in mice and humans). As the field has come to the realization that AD needs to be treated as early as possible, the therapeutic impact these observations remains unclear.
2. A therapeutic approach to counteract these alterations and demonstrating improvement would have been desirable. Unfortunately, the experiment with sorafenib do not carry much weight owing to the lack of the specificity of the drug. Therefore, these observations, while of interest from a cell biological standpoint, offer limited insight into their translational value.
3. This recent paper could be cited to support the notion that myeloid cells in the plaques are microglia (<https://doi.org/10.1084/jem.20191374>).
4. Figure 1 C,D,E: not clear that microglia express less vegf than astrocyte to the extent indicated by the quantification.
5. Figure 1G is not very convincing in terms of integrin expression
6. Please check figure numbering

Reviewer #2 (Remarks to the Author):

The authors have addressed most of my concerns. Given the new data the link to dysregulated Notch signaling is much clearer. I agree that crossing the mice with Notch transgenic would take too long in these times.

There are no further major points.

Minor point:

The authors made some mistakes in referring to the figures (e.g. they mention figure 4 in the text but the information is displayed on fig. 5).

Reviewer #3 (Remarks to the Author):

The authors have done a good job in answering this reviewer's previous critique.

Point-by-point response to the reviewers' commentaries

(Reviewers criticisms have been numerated to facilitate the reading. Original comments are in black and our responses in green. References to new text or figures are showed in red).

Reviewers' comments:

Reviewer #1 (Remarks to the Author):

The authors have done an excellent job at addressing the concerns raised in the review.

We thank the reviewer for her/his kind comments and for the time employed in the evaluation of our manuscript.

The following questions remain:

1. The cell biological phenomena described here occur quite late in the progression of AD (as shown both in mice and humans). As the field has come to the realization that AD needs to be treated as early as possible, the therapeutic impact these observations remains unclear.

We agree with the reviewer that A β plaque deposition is considered a late event in the progression of AD. However, our hope and of many in the field is to reverse the damage caused by A β plaques and by that to cease or at least slowdown the progression of the disease. To make this point clear, we have added a paragraph at the end of the discussion section (Page 23, lines 20 to 25).

2. A therapeutic approach to counteract these alterations and demonstrating improvement would have been desirable. Unfortunately, the experiment with sorafenib do not carry much weight owing to the lack of the specificity of the drug. Therefore, these observations, while of interest from a cell biological standpoint, offer limited insight into their translational value.

Following the editor's recommendation, we have now discussed possible alternative treatments at the end of the discussion section (Page 23, lines 20 to 25).

3. This recent paper could be cited to support the notion that myeloid cells in the plaques are microglia (<https://doi.org/10.1084/jem.20191374>).

We thank the reviewer for the suggestion, we have now added the reference in the discussion section (Page 23, lines 5 and 6).

4. Figure 1 C,D,E: not clear that microglia express less vegf than astrocyte to the extent indicated by the quantification.

We have changed the figure to better reflect the quantification of the images (and also changed the figure legend accordingly). We apologize for showing the exception in the figure more than the generality, that is, no expression of *Vegfa* in most ABAM.

5. Figure 1G is not very convincing in terms of integrin expression

We thank the reviewer for the suggestion. We have now changed the image to better show the increase in integrin expression around A β plaques. We have also added a sentence in the description of those results highlighting the changes in morphology of these cells, which suggest vascular remodeling (Page 7, lines 9 and 11).

6. Please check figure numbering

We deeply apologize for the mistakes. We have now thoroughly reviewed Figures' numbering.

Reviewer #2 (Remarks to the Author):

The authors have addressed most of my concerns. Given the new data the link to dysregulated Notch signaling is much clearer. I agree that crossing the mice with Notch transgenic would take too long in these times.

We thank the reviewer for her/his kind comments and for the time employed in the evaluation of our manuscript.

There are no further major points.

Minor point:

The authors made some mistakes in referring to the figures (e.g. they mention figure 4 in the text but the information is displayed on fig. 5).

We sincerely apologize for the mistakes. We have now thoroughly reviewed Figures' numbering.

Reviewer #3 (Remarks to the Author):

The authors have done a good job in answering this reviewer's previous critique.

We thank the reviewer for her/his kind comment and for the time employed in the evaluation of our manuscript.

REVIEWERS' COMMENTS

Reviewer #1 (Remarks to the Author):

No further comments